# Explicit Expressions for Most Common Entropies

**DOI:** 10.3390/e25030534

**Published:** 2023-03-20

**Authors:** Saralees Nadarajah, Malick Kebe

**Affiliations:** Department of Mathematics, Howard University, Washington, DC 20059, USA; malick.kebe@bison.howard.edu

**Keywords:** beta function, gamma function, Gauss hypergeometric function

## Abstract

Entropies are useful measures of variation. However, explicit expressions for entropies available in the literature are limited. In this paper, we provide a comprehensive collection of explicit expressions for four of the most common entropies for over sixty continuous univariate distributions. Most of the derived expressions are new. The explicit expressions involve known special functions.

## 1. Introduction

Let *X* denote a continuous random variable with probability density and cumulative distribution functions specified by $f_{X}\left(\cdot \right)$ and $F_{X}\left(\cdot \right)$, respectively. Four of the most popular entropies are the geometric mean [1,2], Shannon entropy ([3], pp. 379–423; [3], pp. 623–656), Rényi entropy [4] and the cumulative residual entropy [5], defined by
(1)$$\begin{array}{c} GM\left(X\right)=\int \log xf_{X}\left(x\right)dx, \end{array}$$
(2)$$\begin{array}{c} S\left(X\right)=-\int \log f_{X}\left(x\right)f_{X}\left(x\right)dx, \end{array}$$
(3)$$\begin{array}{c} R\left(X\right)=\frac{1}{1-\gamma }\log\left\{\int {\left[f_{X}\left(x\right)\right]}^{\gamma }dx\right\} \end{array}$$
and
(4)$$\begin{array}{c} CE\left(X\right)=-\int \left[1-F_{X}\left(x\right)\right]\log\left[1-F_{X}\left(x\right)\right]dx, \end{array}$$
respectively, for γ≥0 and γ≠1.

There have been several papers giving explicit expressions for entropies. Ref. [6] derived expressions for S(X) for twenty univariate distributions. Ref. [7] derived expressions for S(X) for five multivariate distributions. Ref. [8] derived expressions for S(X) and mutual information for eight multivariate distributions. Ref. [9] derived expressions for S(X) and R(X) for fifteen bivariate distributions. Ref. [10] derived expressions for S(X) and R(X) for fifteen multivariate distributions. Ref. [11] derived expressions for S(X), R(X) and the *q*-entropy for the Dagum distribution. Ref. [12] derived expressions for S(X) for certain binomial type distributions. Ref. [13] derived expressions for GM(X) and CE(X) for three Lindley type distributions.

All of these and other papers are restrictive in terms of the entropies considered and the number of distributions considered. In this paper, we derive expressions for (Equation 1)–(Equation 4) for more than sixty continuous univariate distributions, see Section 3. Most of the derived expressions are new. Some technicalities used in the derivations are given in Section 2. The derivations themselves are not given and can be obtained from the corresponding author. Some conclusions and future work are noted in Section 4.

The calculations of this paper involve several special functions, including the the exponential integral defined by
$$\begin{array}{c} \mathrm{Ei}\left(a\right)=\int_{-\infty }^{a}\frac{\exp(t)}{t}dt; \end{array}$$
the gamma function defined by
$$\begin{array}{c} \Gamma \left(a\right)=\int_{0}^{\infty }t^{a-1}\exp\left(-t\right)dt; \end{array}$$
the lower incomplete gamma function defined by
$$\begin{array}{c} \Gamma \left(a,x\right)=\int_{x}^{\infty }t^{a-1}\exp\left(-t\right)dt; \end{array}$$
the upper incomplete gamma function defined by
$$\begin{array}{c} \gamma \left(a,x\right)=\int_{0}^{x}t^{a-1}\exp\left(-t\right)dt; \end{array}$$
the digamma function defined by
$$\begin{array}{c} \psi \left(a\right)=\frac{\log\Gamma (a)}{da}; \end{array}$$
the standard normal distribution function defined by
$$\begin{array}{c} \Phi \left(a\right)=\frac{1}{\sqrt{2\pi }}\int_{-\infty }^{x}\exp\left(-\frac{t^{2}}{2}\right)dt; \end{array}$$
the error function defined by
$$\begin{array}{c} \mathrm{erf}\left(a\right)=\frac{2}{\sqrt{\pi }}\int_{0}^{a}\exp\left(-t^{2}\right)dt; \end{array}$$
the complementary error function defined by
$$\begin{array}{c} \mathrm{erfc}\left(a\right)=\frac{2}{\sqrt{\pi }}\int_{a}^{\infty }\exp\left(-t^{2}\right)dt; \end{array}$$
the beta function defined by
$$\begin{array}{c} B\left(a,b\right)=\int_{0}^{1}t^{a-1}{\left(1-t\right)}^{b-1}dt; \end{array}$$
the incomplete beta function defined by
$$\begin{array}{c} B_{x}\left(a,b\right)=\int_{0}^{x}t^{a-1}{\left(1-t\right)}^{b-1}dt; \end{array}$$
the incomplete beta function ratio defined by
$$\begin{array}{c} I_{x}\left(a,b\right)=\frac{B_{x}\left(a,b\right)}{B(a,b)}; \end{array}$$
the modified Bessel function of the first kind of order ν defined by
$$\begin{array}{c} I_{\nu }\left(x\right)=\sum_{k=0}^{\infty }\frac{1}{\Gamma (k+\nu +1)k!}{\left(\frac{x}{2}\right)}^{2k+\nu }; \end{array}$$
the modified Bessel function of the second kind defined by
$$\begin{array}{c} K_{\nu }\left(x\right)=\left\{\begin{array}{cc} \frac{\pi }{2\sin(\pi \nu )}\left[I_{-\nu }\left(x\right)-I_{\nu }\left(x\right)\right], & \mathrm{if}\;\nu \notin Z,\\ \lim_{\mu \to \nu }K_{\mu }\left(x\right), & \mathrm{if}\;\nu \in Z; \end{array}\right. \end{array}$$
the confluent hypergeometric function defined by
$$\begin{array}{c} {}_{1}F_{1}\left(a;b;x\right)=\sum_{k=0}^{\infty }\frac{{\left(a\right)}_{k}}{{\left(b\right)}_{k}}\frac{x^{k}}{k!}, \end{array}$$
where ${\left(a\right)}_{k}=a\left(a+1\right)\cdots \left(a+k-1\right)$ denotes the ascending factorial; the Kummer function defined by
$$\begin{array}{c} \Psi \left(a;b;x\right)=\frac{\Gamma (1-b)}{\Gamma (1+a-b)}\;{}_{1}F_{1}\left(a;b;x\right)+\frac{\Gamma (b-1)}{\Gamma (a)}x^{1-b}\;{}_{1}F_{1}\left(1+a-b;1-b;x\right); \end{array}$$
the Gauss hypergeometric function defined by
$$\begin{array}{c} {}_{2}F_{1}\left(a,b;c;x\right)=\sum_{k=0}^{\infty }\frac{{\left(a\right)}_{k}{\left(b\right)}_{k}}{{\left(c\right)}_{k}}\frac{x^{k}}{k!}; \end{array}$$
the degenerate hypergeometric series of two variables defined by
$$\begin{array}{c} \Phi_{1}\left(a,b,c,x,y\right)=\sum_{m=0}^{\infty }\sum_{n=0}^{\infty }\frac{{\left(a\right)}_{m+n}{\left(b\right)}_{n}x^{m}y^{n}}{{\left(c\right)}_{m+n}m!n!}; \end{array}$$
the degenerate hypergeometric function of two variables defined by
$$\begin{array}{c} F_{1}\left(a,b,c;d;x,y\right)=\sum_{m=0}^{\infty }\sum_{n=0}^{\infty }\frac{{\left(a\right)}_{m+n}{\left(b\right)}_{m}{\left(c\right)}_{n}x^{m}y^{n}}{{\left(d\right)}_{m+n}m!n!}. \end{array}$$

The properties of these special functions can be found in [14,15].

## 2. Technical Lemmas

The derivations in Section 3 use the following two lemmas.

**Lemma** **1.**
*The geometric mean defined by (Equation 1) can be calculated using*

(5)
$$\begin{array}{c} GM\left(X\right)={\left.\frac{d}{d\alpha }E\left(X^{\alpha }\right)\right|}_{\alpha =0}, \end{array}$$

*where E(·) denotes the expectation defined by*

$$\begin{array}{c} E\left(X^{\alpha }\right)=\int x^{\alpha }f_{X}\left(x\right)dx. \end{array}$$



**Proof.** Note that
$$\begin{array}{c} GM\left(X\right)=\int {\left.\frac{d}{d\alpha }x^{\alpha }\right|}_{\alpha =0}f_{X}\left(x\right)dx={\left.\frac{d}{d\alpha }\left[\int x^{\alpha }f_{X}\left(x\right)dx\right]\right|}_{\alpha =0}. \end{array}$$Hence, the result. □

**Lemma** **2.**
*The cumulative residual entropy defined by (Equation 4) can be calculated using*

(6)
$$\begin{array}{c} CE\left(X\right)=\sum_{k=1}^{\infty }\frac{1}{k}\int {\left[F_{X}\left(x\right)\right]}^{k}dx-\sum_{k=1}^{\infty }\frac{1}{k}\int {\left[F_{X}\left(x\right)\right]}^{k+1}dx. \end{array}$$



**Proof.** Using the Taylor series expansion for log(1−z), we can write
$$\begin{array}{c} CE\left(X\right)=\int \left[1-F_{X}\left(x\right)\right]\sum_{k=1}^{\infty }\frac{1}{k}{\left[F_{X}\left(x\right)\right]}^{k}dx=\int \sum_{k=1}^{\infty }\frac{1}{k}{\left[F(x)\right]}^{k}dx-\int \sum_{k=1}^{\infty }\frac{1}{k}{\left[F(x)\right]}^{k+1}dx. \end{array}$$Hence, the result. □

## 3. The Tabulation

In this section, we give expressions for $f_{X}\left(x\right)$ (the probability density function), $F_{X}\left(x\right)$ (the cumulative distribution function), GM(X) (the geometric mean), S(X) (Shannon entropy), R(X) (Rényi entropy), and CE(X) (the cumulative residual entropy) for over sixty continuous univariate distributions.

1. Gauss hypergeometric beta distribution [16]: for this distribution,
$$\begin{array}{c} f_{X}\left(x\right)=\frac{Kx^{a-1}{\left(1-x\right)}^{b-1}}{{\left(1+dx\right)}^{c}}, \end{array}$$
$$\begin{array}{c} F_{X}\left(x\right)=\frac{Kx^{a}}{a}F_{1}\left(a,c,1-b,a+1;-dx,x\right), \end{array}$$
$$\begin{array}{c} GM\left(X\right)=\exp\left[\frac{\Gamma^{{}^{\prime }}\left(a\right)}{\Gamma (a)}+{\left.\frac{1}{{}_{2}F_{1}\left(c,a;a+b;-d\right)}\frac{\partial }{\partial \alpha }\;{}_{2}F_{1}\left(c,\alpha +a;\alpha +a+b;-d\right)\right|}_{\alpha =0}-\frac{\Gamma^{{}^{\prime }}\left(a+b\right)}{\Gamma (a+b)}\right], \end{array}$$
$$\begin{array}{ccc} & & S\left(X\right)=-\frac{\Gamma^{{}^{\prime }}\left(a\right)}{\Gamma (a)}-{\left.\frac{1}{{}_{2}F_{1}\left(c,a;a+b;-d\right)}\frac{\partial }{\partial \alpha }\;{}_{2}F_{1}\left(c,\alpha +a;\alpha +a+b;-d\right)\right|}_{\alpha =0}\\ & & \;\;+\frac{2\Gamma^{{}^{\prime }}\left(a+b\right)}{\Gamma (a+b)}-\frac{\Gamma^{{}^{\prime }}\left(b\right)}{\Gamma (b)}\\ & & \;\;-{\left.\frac{1}{{}_{2}F_{1}\left(c,a;a+b;-d\right)}\frac{\partial }{\partial \alpha }\;{}_{2}F_{1}\left(c,a;\alpha +a+b;-d\right)\right|}_{\alpha =0}\\ & & \;\;-\frac{1}{{}_{2}F_{1}\left(c,a;a+b;-d\right)}\left.\frac{d}{d\alpha }\;{}_{2}F_{1}\left(c-\alpha ,a;a+b;-d\right)\right.-\log B\left(a,b\right)\\ & & \;\;-\log\;{}_{2}F_{1}\left(c-\alpha ,a;a+b;-d\right), \end{array}$$
$$\begin{array}{ccc} & & R\left(X\right)=\frac{1}{1-\gamma }\log B\left(a\gamma -\gamma +1,b\gamma -\gamma +1\right)\\ & & \;\;+\frac{1}{1-\gamma }\log\;{}_{2}F_{1}\left(c\gamma ,a\gamma -\gamma +1;a\gamma +b\gamma -2\gamma +2;-d\right)\\ & & \;\;-\frac{\gamma }{1-\gamma }\log B\left(a,b\right)-\frac{\gamma }{1-\gamma }\log\;{}_{2}F_{1}\left(c,a;a+b;-d\right) \end{array}$$
and
$$\begin{array}{c} CE\left(X\right)=-\int_{0}^{1}\left[1-\frac{Kx^{a}}{a}F_{1}\left(a,c,1-b,a+1;-dx,x\right)\right]\log\left[1-\frac{Kx^{a}}{a}F_{1}\left(a,c,1-b,a+1;-dx,x\right)\right]dx \end{array}$$
for 0<x<1, a>0, b>0, −∞<c<∞ and d>−1, where $\Gamma^{{}^{\prime }}\left(x\right)=\frac{d\Gamma (x)}{dx}$ and $\frac{1}{K}=B\left(a,b\right)\;{}_{2}F_{1}\left(c,a;a+b;-d\right)$.

2. *q* Weibull distribution [17]: for this distribution,
$$\begin{array}{c} f_{X}\left(x\right)=\left(2-q\right)abx^{a-1}{\left[1-\left(1-q\right)bx^{a}\right]}^{\frac{1}{1-q}}, \end{array}$$
$$\begin{array}{c} F_{X}\left(x\right)=1-{\left[1-\left(1-q\right)bx^{a}\right]}^{\frac{2-q}{1-q}}, \end{array}$$
$$\begin{array}{c} GM\left(X\right)=\left\{\begin{array}{cc} -\frac{2-q}{a(q-1)}\log\left[(q-1)b\right]\frac{\Gamma \left(\frac{2-q}{q-1}\right)}{\Gamma \left(\frac{1}{q-1}\right)}-\frac{2-q}{a(q-1)}\frac{\Gamma^{{}^{\prime }}\left(\frac{2-q}{q-1}\right)}{\Gamma \left(\frac{1}{q-1}\right)}\\ \;+\frac{2-q}{a(q-1)}\frac{\Gamma^{{}^{\prime }}\left(1\right)\Gamma \left(\frac{2-q}{q-1}\right)}{\Gamma \left(\frac{1}{q-1}\right)}, & \mathrm{if}\;1<q<2,\\ \frac{2-q}{a(q-1)}\log\left[(1-q)b\right]\frac{\Gamma \left(\frac{2-q}{1-q}\right)}{\Gamma \left(\frac{3-2q}{1-q}\right)}+\frac{2-q}{a(q-1)}\frac{\Gamma^{{}^{\prime }}\left(\frac{3-2q}{1-q}\right)}{{\left[\Gamma \left(\frac{3-2q}{1-q}\right)\right]}^{2}}\\ \;-\frac{2-q}{a(q-1)}\frac{\Gamma^{{}^{\prime }}\left(1\right)\Gamma \left(\frac{2-q}{1-q}\right)}{\Gamma \left(\frac{3-2q}{1-q}\right)}, & \mathrm{if}\;q<1, \end{array}\right. \end{array}$$
$$\begin{array}{c} S\left(X\right)=\left\{\begin{array}{cc} -\log\left[(2-q)ab\right]+\frac{1}{2-q}-\frac{\left(1-a)(2-q\right)}{a(q-1)}\log\left[(q-1)b\right]\frac{\Gamma \left(\frac{2-q}{q-1}\right)}{\Gamma \left(\frac{1}{q-1}\right)}\\ \;-\frac{(1-a)2-q}{a(q-1)}\frac{\Gamma^{{}^{\prime }}\left(\frac{2-q}{q-1}\right)}{\Gamma \left(\frac{1}{q-1}\right)}\\ \;+\frac{\left(1-a)(2-q\right)}{a(q-1)}\frac{\Gamma^{{}^{\prime }}\left(1\right)\Gamma \left(\frac{2-q}{q-1}\right)}{\Gamma \left(\frac{1}{q-1}\right)}, & \mathrm{if}\;1<q<2,\\ -\log\left[(2-q)ab\right]+\frac{1}{2-q}+\frac{\left(1-a)(2-q\right)}{a(q-1)}\log\left[(1-q)b\right]\frac{\Gamma \left(\frac{2-q}{1-q}\right)}{\Gamma \left(\frac{3-2q}{1-q}\right)}\\ \;+\frac{\left(1-a)(2-q\right)}{a(q-1)}\frac{\Gamma^{{}^{\prime }}\left(\frac{3-2q}{1-q}\right)}{{\left[\Gamma \left(\frac{3-2q}{1-q}\right)\right]}^{2}}\\ \;-\frac{\left(1-a)(2-q\right)}{a(q-1)}\frac{\Gamma^{{}^{\prime }}\left(1\right)\Gamma \left(\frac{2-q}{1-q}\right)}{\Gamma \left(\frac{3-2q}{1-q}\right)}, & \mathrm{if}\;q<1, \end{array}\right. \end{array}$$
$$\begin{array}{c} R\left(X\right)=\left\{\begin{array}{cc} \frac{1}{1-\gamma }\log\left[\frac{{\left(2-q\right)}^{\gamma }a^{\gamma -1}b^{\frac{\gamma -1}{a}}}{{\left(q-1\right)}^{\gamma +\frac{1-\gamma }{a}}}B\left(\gamma +\frac{1-\gamma }{a},-\frac{\gamma }{1-q}-\gamma -\frac{1-\gamma }{a}\right)\right], & \mathrm{if}\;1<q<2,\\ \log\left[\frac{{\left(2-q\right)}^{\gamma }a^{\gamma -1}b^{\frac{\gamma -1}{a}}}{{\left(q-1\right)}^{\gamma +\frac{1-\gamma }{a}}}B\left(\gamma +\frac{1-\gamma }{a},1+\frac{\gamma }{1-q}\right)\right], & \mathrm{if}\;q<1 \end{array}\right. \end{array}$$
and
$$\begin{array}{c} CE\left(X\right)=\left\{\begin{array}{cc} -\frac{2-q}{a(q-1)}\frac{\Gamma \left(\frac{1}{a}\right)}{{\left[(q-1)b\right]}^{\frac{1}{a}}}\frac{\Gamma^{{}^{\prime }}\left(\frac{2-q}{q-1}-\frac{1}{a}\right)}{\Gamma \left(\frac{2-q}{q-1}\right)}\\ \;+\frac{2-q}{a(q-1)}\frac{\Gamma \left(\frac{1}{a}\right)}{{\left[(q-1)b\right]}^{\frac{1}{a}}}\frac{\Gamma \left(\frac{2-q}{q-1}-\frac{1}{a}\right)\Gamma^{{}^{\prime }}\left(\frac{2-q}{q-1}\right)}{{\left[\Gamma \left(\frac{2-q}{q-1}\right)\right]}^{2}}, & \mathrm{if}\;1<q<2,\\ \frac{2-q}{a(q-1)}\frac{\Gamma \left(\frac{1}{a}\right)}{{\left[(1-q)b\right]}^{\frac{1}{a}}}\frac{\Gamma^{{}^{\prime }}\left(\frac{2-q}{1-q}+1\right)}{\Gamma \left(\frac{2-q}{1-q}+1+\frac{1}{a}\right)}\\ \;-\frac{2-q}{a(q-1)}\frac{\Gamma \left(\frac{1}{a}\right)}{{\left[(1-q)b\right]}^{\frac{1}{a}}}\frac{\Gamma \left(\frac{2-q}{1-q}+1\right)\Gamma^{{}^{\prime }}\left(\frac{2-q}{1-q}+1+\frac{1}{a}\right)}{{\left[\Gamma \left(\frac{2-q}{1-q}+1+\frac{1}{a}\right)\right]}^{2}}, & \mathrm{if}\;q<1 \end{array}\right. \end{array}$$
for a>0, b>0, 0<x<∞ if 1<q<2 and $0<x<{\left[(1-q)b\right]}^{\frac{1}{a}}$ if q<1.

3. *q* exponential distribution [17]: for this distribution,
$$\begin{array}{c} f_{X}\left(x\right)=\left(2-q\right)b{\left[1-(1-q)bx\right]}^{\frac{1}{1-q}}, \end{array}$$
$$\begin{array}{c} F_{X}\left(x\right)=1-{\left[1-(1-q)bx\right]}^{\frac{2-q}{1-q}}, \end{array}$$
$$\begin{array}{c} GM\left(X\right)=\left\{\begin{array}{cc} -\frac{2-q}{q-1}\frac{\Gamma^{{}^{\prime }}\left(\frac{q}{q-1}\right)}{\Gamma \left(\frac{2q-1}{q-1}\right)}+\frac{2-q}{q}\Gamma^{{}^{\prime }}\left(1\right)-\frac{2-q}{q}\log\left[(q-1)b\right], & \mathrm{if}\;1<q<2,\\ -\frac{\Gamma^{{}^{\prime }}\left(\frac{3-2q}{1-q}\right)}{\Gamma \left(\frac{3-2q}{1-q}\right)}+\Gamma^{{}^{\prime }}\left(1\right)-\log\left[(1-q)b\right], & \mathrm{if}\;q<1, \end{array}\right. \end{array}$$
$$\begin{array}{c} S\left(X\right)=-\log\left[(2-q)b\right]-\frac{2-q}{1-q}\frac{1}{1+(\alpha +1)(1-q)}, \end{array}$$
$$\begin{array}{c} R\left(X\right)=-\log b+\frac{1}{1-\gamma }\log\frac{{\left(2-q\right)}^{\gamma }}{\gamma +1-q} \end{array}$$
and
$$\begin{array}{c} CE\left(X\right)=\frac{2-q}{b{\left(3-2q\right)}^{2}} \end{array}$$
for b>0, 0<x<∞ if 1<q<2 and 0<x<(1−q)b if q<1.

4. Weighted exponential distribution: for this distribution,
$$\begin{array}{c} f_{X}\left(x\right)=\frac{a+1}{a}b\exp\left(-bx\right)\left[1-\exp(-abx)\right], \end{array}$$
$$\begin{array}{c} F_{X}\left(x\right)=\frac{1}{a}\exp\left(-bx\right)\left[\exp(-abx)-a-1\right], \end{array}$$
$$\begin{array}{c} GM\left(X\right)=a\Gamma^{{}^{\prime }}\left(1\right)-a\log b+\left(a+1\right)\log\left(a+1\right), \end{array}$$
$$\begin{array}{c} S\left(X\right)=-\log\frac{(a+1)b}{a}+a+1-\frac{1}{a+1}-\Gamma^{{}^{\prime }}\left(2\right)+\frac{\Gamma^{{}^{\prime }}\left(\frac{1}{a}+2\right)}{\Gamma \left(\frac{1}{a}+2\right)}, \end{array}$$
$$\begin{array}{c} R\left(X\right)=-\log b-\frac{1+\gamma }{1-\gamma }\log a+\frac{\gamma }{1-\gamma }\log\left(a+1\right)+\frac{1}{1-\gamma }\log B\left(\frac{\gamma }{a},\gamma +1\right) \end{array}$$
and
$$\begin{array}{c} CE\left(X\right)=\frac{a+1}{ab}\left[1-\frac{1}{{\left(1+a\right)}^{3}}\right]+\frac{(a+1)\log a}{ab}\left[1-\frac{1}{{\left(1+a\right)}^{2}}\right] \end{array}$$
for x>0, a>0 and b>0.

5. Teissier distribution [18]: for this distribution,
$$\begin{array}{c} f_{X}\left(x\right)=\left[\exp(ax)-1\right]\exp\left[ax-\exp(ax)+1\right], \end{array}$$
$$\begin{array}{c} F_{X}\left(x\right)=1-\exp\left[ax-\exp(ax)+1\right], \end{array}$$
$$\begin{array}{c} GM\left(X\right)=e{\left.\frac{\partial }{\partial \alpha }\left[a^{-\alpha }\int_{1}^{\infty }{\left(\log y\right)}^{\alpha }\left(y-1\right)\exp\left(-y\right)dy\right]\right|}_{\alpha =0}, \end{array}$$
$$\begin{array}{c} S\left(X\right)=-\log a-ae{\left.\frac{\partial }{\partial \alpha }\int_{1}^{\infty }y{\left(y-1\right)}^{\alpha +1}\exp\left(-y\right)dy\right|}_{\alpha =0}-e{\left.\frac{\partial }{\partial \alpha }\left[\Gamma (\alpha +2,1)-\Gamma (\alpha +1,1)\right]\right|}_{\alpha =0}+2, \end{array}$$
$$\begin{array}{c} R\left(X\right)=a^{\gamma }\Gamma \left(\gamma +1\right)\Psi \left(\gamma +2,2\gamma +1;\gamma \right) \end{array}$$
and
$$\begin{array}{c} CE\left(X\right)=-\frac{e}{a}\left[\mathrm{Ei}(1)-\exp(-1)\right] \end{array}$$
for x>0 and a>0.

6. Maxwell distribution [19,20]: for this distribution,
$$\begin{array}{c} f_{X}\left(x\right)=\frac{4a^{\frac{3}{2}}}{\sqrt{\pi }}x^{2}\exp\left(-ax^{2}\right), \end{array}$$
$$\begin{array}{c} F_{X}\left(x\right)=\frac{2}{\sqrt{\pi }}\gamma \left(\frac{3}{2},ax^{2}\right), \end{array}$$
$$\begin{array}{c} GM\left(X\right)=\frac{1-\log a}{2}, \end{array}$$
$$\begin{array}{c} S\left(X\right)=\frac{-\Gamma \left(\frac{3}{2}\right)\log a+\Gamma^{{}^{\prime }}\left(\frac{3}{2}\right)}{\sqrt{\pi }}, \end{array}$$
$$\begin{array}{c} R\left(X\right)=-\log\left(2\sqrt{a}\right)+\frac{\log2}{1-\gamma }-\frac{\gamma \log\pi }{2(1-\gamma )}-\left(\gamma +\frac{1}{2}\right)\frac{\log\gamma }{1-\gamma }+\frac{1}{1-\gamma }\log\Gamma \left(\gamma +\frac{1}{2}\right) \end{array}$$
and
$$\begin{array}{c} CE\left(X\right)=-\frac{2}{\sqrt{\pi }}\int_{0}^{\infty }\left[\log2-\log\sqrt{\pi }+\log\Gamma \left(\frac{3}{2},ax^{2}\right)\right]\Gamma \left(\frac{3}{2},ax^{2}\right)dx \end{array}$$
for x>0 and a>0.

7. Inverse Maxwell distribution: for this distribution,
$$\begin{array}{c} f_{X}\left(x\right)=\frac{4a^{\frac{3}{2}}}{\sqrt{\pi }}x^{-4}\exp\left(-ax^{-2}\right), \end{array}$$
$$\begin{array}{c} F_{X}\left(x\right)=\frac{2}{\sqrt{\pi }}\Gamma \left(\frac{3}{2},ax^{-2}\right), \end{array}$$
$$\begin{array}{c} GM\left(X\right)=\frac{\Gamma \left(\frac{3}{2}\right)\log a-\Gamma^{{}^{\prime }}\left(\frac{3}{2}\right)}{\sqrt{\pi }}, \end{array}$$
$$\begin{array}{c} S\left(X\right)=-\log\left(\frac{4}{\sqrt{\pi }}\right)+\frac{1}{2}\log a-\frac{4}{\sqrt{\pi }}\Gamma^{{}^{\prime }}\left(\frac{3}{2}\right), \end{array}$$
$$\begin{array}{c} R\left(X\right)=\frac{1}{1-\gamma }\log\frac{2a^{\frac{5}{2}}}{\sqrt{\pi }}+\frac{\gamma }{1-\gamma }\log\frac{4}{a\sqrt{\pi }}+\frac{1}{1-\gamma }\log\Gamma \left(2\gamma -\frac{1}{2}\right) \end{array}$$
and
$$\begin{array}{c} CE\left(X\right)=-\frac{2\sqrt{a}}{\sqrt{\pi }}\log\frac{2}{\sqrt{\pi }}-\frac{2}{\sqrt{\pi }}\int_{0}^{\infty }\gamma \left(\frac{3}{2},ax^{-2}\right)\log\gamma \left(\frac{3}{2},ax^{-2}\right)dx \end{array}$$
for x>0 and a>0.

8. Power Maxwell distribution [21]: for this distribution,
$$\begin{array}{c} f_{X}\left(x\right)=\frac{4ab^{\frac{3}{2}}}{\sqrt{\pi }}x^{3a-1}\exp\left(-bx^{2a}\right), \end{array}$$
$$\begin{array}{c} F_{X}\left(x\right)=\frac{2}{\sqrt{\pi }}\gamma \left(\frac{3}{2},bx^{2a}\right), \end{array}$$
$$\begin{array}{c} GM\left(X\right)=\frac{-\Gamma \left(\frac{3}{2}\right)\log b+\Gamma^{{}^{\prime }}\left(\frac{3}{2}\right)}{a\sqrt{\pi }}, \end{array}$$
$$\begin{array}{c} S\left(X\right)=\frac{3}{2}-\frac{\log b}{2a}+\frac{1-3a}{a\sqrt{\pi }}\Gamma^{{}^{\prime }}\left(\frac{3}{2}\right)-\log\frac{4a}{\sqrt{\pi }}, \end{array}$$
$$\begin{array}{c} R\left(X\right)=-\log\left[2ab^{\frac{1}{2a}}\right]+\frac{\log2}{1-\gamma }-\frac{\gamma \log\pi }{2(1-\gamma )}-\left(\frac{3\gamma }{2}+\frac{1-\gamma }{2a}\right)\frac{\log\gamma }{1-\gamma }+\frac{1}{1-\gamma }\log\Gamma \left(\frac{3\gamma }{2}+\frac{1-\gamma }{2a}\right) \end{array}$$
and
$$\begin{array}{c} CE\left(X\right)=-\frac{2}{\sqrt{\pi }b^{\frac{1}{2a}}}\log\left(\frac{2}{\sqrt{\pi }}\right)\Gamma \left(\frac{1}{2a}+\frac{3}{2}\right)-\frac{2}{\sqrt{\pi }}\int_{0}^{\infty }\gamma \left(\frac{3}{2},bx^{2a}\right)\log\Gamma \left(\frac{3}{2},bx^{2a}\right)dx \end{array}$$
for x>0, a>0 and b>0.

9. Inverse power Maxwell distribution [22]: for this distribution,
$$\begin{array}{c} f_{X}\left(x\right)=\frac{4ab^{\frac{3}{2}}}{\sqrt{\pi }}x^{-3a-1}\exp\left(-bx^{-2a}\right), \end{array}$$
$$\begin{array}{c} F_{X}\left(x\right)=\frac{2}{\sqrt{\pi }}\Gamma \left(\frac{3}{2},bx^{-2a}\right), \end{array}$$
$$\begin{array}{c} GM\left(X\right)=\frac{\Gamma \left(\frac{3}{2}\right)\log b-\Gamma^{{}^{\prime }}\left(\frac{3}{2}\right)}{a\sqrt{\pi }}, \end{array}$$
$$\begin{array}{c} S\left(X\right)=-\log\left(\frac{4a}{\sqrt{\pi }}\right)+\frac{1}{2a}\log b-\frac{3a+1}{a\sqrt{\pi }}\Gamma^{{}^{\prime }}\left(\frac{3}{2}\right), \end{array}$$
$$\begin{array}{c} R\left(X\right)=\frac{1}{1-\gamma }\log\frac{2b^{\frac{5}{2}}}{\sqrt{\pi }}+\frac{\gamma }{1-\gamma }\log\frac{4a}{b\sqrt{\pi }}+\frac{1}{1-\gamma }\log\Gamma \left(\frac{3\gamma }{2}+\frac{\gamma -1}{2a}\right) \end{array}$$
and
$$\begin{array}{c} CE\left(X\right)=-\frac{2}{\sqrt{\pi }}b^{\frac{1}{2a}}\log\left(\frac{2}{\sqrt{\pi }}\right)\Gamma \left(\frac{3}{2}-\frac{1}{2a}\right)-\frac{2}{\sqrt{\pi }}\int_{0}^{\infty }\gamma \left(\frac{3}{2},bx^{-2a}\right)\log\gamma \left(\frac{3}{2},bx^{-2a}\right)dx \end{array}$$
for x>0, a>0 and b>0.

10. Omega distribution [23]: for this distribution,
$$\begin{array}{c} f_{X}\left(x\right)=\frac{abx^{b-1}}{1-x^{2b}}{\left(\frac{1+x^{b}}{1-x^{b}}\right)}^{-\frac{a}{2}}, \end{array}$$
$$\begin{array}{c} F_{X}\left(x\right)=1-{\left(\frac{1+x^{b}}{1-x^{b}}\right)}^{-\frac{a}{2}}, \end{array}$$
$$\begin{array}{c} GM\left(X\right)=a{\left.\frac{\partial }{\partial \alpha }\left[B\left(\frac{\alpha }{b}+1,\frac{a}{2}\right)\;{}_{2}F_{1}\left(\frac{\alpha }{b}+1,\frac{a}{2}+1;\frac{\alpha }{b}+1+\frac{a}{2};-1\right)\right]\right|}_{\alpha =0}, \end{array}$$
$$\begin{array}{ccc} & & S\left(X\right)=-\log\left(ab\right)+a\left(1-b\right){\left.\frac{\partial }{\partial \alpha }\left[B\left(\frac{\alpha }{b}+1,\frac{a}{2}\right)\;{}_{2}F_{1}\left(\frac{\alpha }{b}+1,\frac{a}{2}+1;\frac{\alpha }{b}+1+\frac{a}{2};-1\right)\right]\right|}_{\alpha =0}\\ & & \;\;+\left(a+2\right){\left.\frac{\partial }{\partial \alpha }\;{}_{2}F_{1}\left(1,\frac{a}{2}+1-\alpha ;\frac{\alpha }{2}+1;-1\right)\right|}_{\alpha =0}\\ & & \;\;-a\left(a-2\right){\left.\frac{\partial }{\partial \alpha }\left[\frac{1}{a+2\alpha }\;{}_{2}F_{1}\left(1,\frac{a}{2}+1;\alpha +1+\frac{a}{2};-1\right)\right]\right|}_{\alpha =0}, \end{array}$$
$$\begin{array}{c} R\left(X\right)=\frac{1}{1-\gamma }\log\left[a^{\gamma -1}b^{\gamma }B\left(\gamma +\frac{1-\gamma }{b},1-\gamma +\frac{a\gamma }{2}\right)\;{}_{2}F_{1}\left(\gamma +\frac{1-\gamma }{b},\frac{\gamma a}{2}+\gamma ;\frac{a\gamma }{2}+\frac{1-\gamma }{b}+1;-1\right)\right] \end{array}$$
and
$$\begin{array}{ccc} & & CE\left(X\right)=\frac{a}{2b}{\left.\frac{\partial }{\partial \alpha }\left[B\left(b,\frac{a}{2}+1\right)\;{}_{2}F_{1}\left(b,\frac{a}{2}-\alpha ;b+\frac{a}{2}+1;-1\right)\right]\right|}_{\alpha =0}\\ & & \;\;-\frac{a}{2b}{\left.\frac{\partial }{\partial \alpha }\left[B\left(b,\alpha +\frac{a}{2}+1\right)\;{}_{2}F_{1}\left(b,\frac{a}{2};b+\alpha +\frac{a}{2}+1;-1\right)\right]\right|}_{\alpha =0} \end{array}$$
for x>0, a>0 and b>0.

11. Colak et al.’s distribution [24]: for this distribution,
$$\begin{array}{c} f_{X}\left(x\right)=\frac{a\left(b+1\right){\left(1-x\right)}^{a-1}}{{\left(1+bx\right)}^{a+1}}, \end{array}$$
$$\begin{array}{c} F_{X}\left(x\right)={\left(\frac{1-x}{1+bx}\right)}^{a}, \end{array}$$
$$\begin{array}{c} GM\left(X\right)=a\left(b+1\right){\left.\frac{\partial }{\partial \alpha }\left[B\left(\alpha +1,a\right)\;{}_{2}F_{1}\left(\alpha +1,a+1;\alpha +1+a;-b\right)\right]\right|}_{\alpha =0}, \end{array}$$
$$\begin{array}{ccc} & & S\left(X\right)=-\log\left[a(b+1)\right]+a\left(1-a\right)\left(b+1\right){\left.\frac{\partial }{\partial \alpha }\left[\frac{1}{\alpha +a}\;{}_{2}F_{1}\left(1,a+1;\alpha +a+1;-b\right)\right]\right|}_{\alpha =0}\\ & & \;\;+\left(a+1\right)\left(b+1\right){\left.\frac{\partial }{\partial \alpha }\;{}_{2}F_{1}\left(a,a+1-\alpha ;a+1;-b\right)\right|}_{\alpha =0}, \end{array}$$
$$\begin{array}{c} R\left(X\right)=\frac{1}{1-\gamma }\log\left\{\frac{{\left[a(b+1)\right]}^{\gamma }}{\gamma a-\gamma +1}\;{}_{2}F_{1}\left(1,\gamma a+\gamma ;\gamma a-\gamma +2;-b\right)\right\} \end{array}$$
and
$$\begin{array}{c} CE\left(X\right)=\frac{a}{a+1}{\left.\frac{\partial }{\partial \alpha }\;{}_{2}F_{1}\left(1,a-\alpha ;a+2;-b\right)\right|}_{\alpha =0}-{\left.\frac{\partial }{\partial \alpha }\left[\frac{a}{a+\alpha +1}\;{}_{2}F_{1}\left(1,a;a+\alpha +2;-b\right)\right]\right|}_{\alpha =0} \end{array}$$
for x>0, a>0 and b>0.

12. Bimodal beta distribution [25]: for this distribution,
$$\begin{array}{c} f_{X}\left(x\right)=\frac{\rho +{\left(1-\delta x\right)}^{2}}{CB(\alpha ,\beta )}x^{\alpha -1}{\left(1-x\right)}^{\beta -1}, \end{array}$$
$$\begin{array}{c} F_{X}\left(x\right)=\frac{1}{C}\left[\left(1+\rho \right)I_{x}\left(\alpha ,\beta \right)-2\delta \frac{B_{x}\left(\alpha +1,\beta \right)}{B_{x}\left(\alpha ,\beta \right)}+\delta^{2}\frac{B_{x}\left(\alpha +2,\beta \right)}{B_{x}\left(\alpha ,\beta \right)}\right], \end{array}$$
$$\begin{array}{c} GM\left(X\right)=\frac{\Gamma (\alpha +\beta )}{C\Gamma (\alpha )}\sum_{i=0}^{2}c_{i}\left\{\frac{\Gamma^{{}^{\prime }}\left(\alpha +i\right)}{\Gamma (\alpha +\beta +i)}-\frac{\Gamma \left(\alpha +i\right)\Gamma^{{}^{\prime }}\left(\alpha +\beta +i\right)}{{\left[\Gamma (\alpha +\beta +i)\right]}^{2}}\right\}, \end{array}$$
$$\begin{array}{ccc} & & S\left(X\right)=\log\left[CB(\alpha ,\beta )\right]+\frac{\left(1-\alpha )\Gamma (\alpha +\beta \right)}{C\Gamma (\alpha )}\sum_{i=0}^{2}c_{i}\left\{\frac{\Gamma^{{}^{\prime }}\left(\alpha +i\right)}{\Gamma (\alpha +\beta +i)}-\frac{\Gamma \left(\alpha +i\right)\Gamma^{{}^{\prime }}\left(\alpha +\beta +i\right)}{{\left[\Gamma (\alpha +\beta +i)\right]}^{2}}\right\}\\ & & \;\;+\frac{\left(1-\beta )\Gamma (\alpha +\beta \right)}{C\Gamma (\alpha )}\sum_{i=0}^{2}c_{i}\left\{\frac{\Gamma \left(\alpha +i\right)\Gamma^{{}^{\prime }}\left(\beta \right)}{\Gamma (\alpha +\beta +i)}-\frac{\Gamma \left(\alpha +i\right)\Gamma \left(\beta \right)\Gamma^{{}^{\prime }}\left(\alpha +\beta +i\right)}{{\left[\Gamma (\alpha +\beta +i)\right]}^{2}}\right\}\\ & & \;\;-{\left.\frac{\partial }{\partial a}\left[\frac{{\left(1+\rho \right)}^{a+1}}{C}F_{1}\left(\alpha ,-a-1,-a-1;\alpha +\beta ;\frac{\delta }{1+i\sqrt{\rho }},\frac{\delta }{1-i\sqrt{\rho }}\right)\right]\right|}_{a=0}, \end{array}$$
$$\begin{array}{c} R\left(X\right)=\frac{B\left(\alpha \gamma -\gamma +1,\beta \gamma -\gamma +1\right){\left(1+\rho \right)}^{\gamma }}{C^{\gamma }{\left[B(\alpha ,\beta )\right]}^{\gamma }}F_{1}\left(\alpha \gamma -\gamma +1,-\gamma ,-\gamma ,\alpha \gamma +\beta \gamma -2\gamma +2;\frac{\delta }{1+i\sqrt{\rho }},\frac{\delta }{1-i\sqrt{\rho }}\right) \end{array}$$
and
$$\begin{array}{ccc} & & CE\left(X\right)=-\int_{0}^{1}\left\{1-\frac{1}{C}\left[\left(1+\rho \right)I_{x}\left(\alpha ,\beta \right)-2\delta \frac{B_{x}\left(\alpha +1,\beta \right)}{B_{x}\left(\alpha ,\beta \right)}+\delta^{2}\frac{B_{x}\left(\alpha +2,\beta \right)}{B_{x}\left(\alpha ,\beta \right)}\right]\right\}\\ & & \;\;\;\;\cdot \log\left\{1-\frac{1}{C}\left[\left(1+\rho \right)I_{x}\left(\alpha ,\beta \right)-2\delta \frac{B_{x}\left(\alpha +1,\beta \right)}{B_{x}\left(\alpha ,\beta \right)}+\delta^{2}\frac{B_{x}\left(\alpha +2,\beta \right)}{B_{x}\left(\alpha ,\beta \right)}\right]\right\}dx \end{array}$$
for 0<x<1, α>0, β>0, ρ≥0 and −∞<δ<∞, where $i=\sqrt{-1}$, $c_{0}=1+\rho$, $c_{1}=-2\delta$, $c_{2}=\delta^{2}$ and $C=1+\rho -2\delta \frac{\alpha }{\alpha +\beta }+\delta^{2}\frac{\alpha (\alpha +1)}{\left(\alpha +\beta )(\alpha +\beta +1\right)}$.

13. Confluent hypergeometric beta distribution [26]: for this distribution,
$$\begin{array}{c} f_{X}\left(x\right)=\frac{x^{a-1}{\left(1-x\right)}^{b-1}\exp\left(-cx\right)}{B\left(a,b\right)\;{}_{1}F_{1}\left(a;a+b;-c\right)}, \end{array}$$
$$\begin{array}{c} F_{X}\left(x\right)=\frac{x^{a}\;\Phi_{1}\left(a,1-b,a+1;x,cx\right)}{aB\left(a,b\right)\;{}_{1}F_{1}\left(a;a+b;-c\right)}, \end{array}$$
$$\begin{array}{c} GM\left(X\right)=\frac{\Gamma (a+b)}{\Gamma \left(a\right)\;{}_{1}F_{1}\left(a;a+b;-c\right)}{\left.\frac{\partial }{\partial \alpha }\left[\frac{\Gamma (a+\alpha )}{\Gamma (a+b+\alpha )}\;{}_{1}F_{1}\left(a+\alpha ;a+b+\alpha ;-c\right)\right]\right|}_{\alpha =0}, \end{array}$$
$$\begin{array}{ccc} & & S\left(X\right)=\frac{\left(1-a)\Gamma (a+b\right)}{\Gamma \left(a\right)\;{}_{1}F_{1}\left(a;a+b;-c\right)}{\left.\frac{\partial }{\partial \alpha }\left[\frac{\Gamma (a+\alpha )}{\Gamma (a+b+\alpha )}\;{}_{1}F_{1}\left(a+\alpha ;a+b+\alpha ;-c\right)\right]\right|}_{\alpha =0}\\ & & \;\;+\frac{\left(1-b)\Gamma (a+b\right)}{\Gamma \left(b\right)\;{}_{1}F_{1}\left(a;a+b;-c\right)}{\left.\frac{\partial }{\partial \alpha }\left[\frac{\Gamma (b+\alpha )}{\Gamma (a+b+\alpha )}\;{}_{1}F_{1}\left(a;a+b+\alpha ;-c\right)\right]\right|}_{\alpha =0}\\ & & \;\;+\frac{ca}{a+b}\frac{{}_{1}F_{1}\left(a+1;a+b+1;-c\right)}{{}_{1}F_{1}\left(a;a+b;-c\right)}+\log B\left(a,b\right)+\log\;{}_{1}F_{1}\left(a;a+b;-c\right), \end{array}$$
$$\begin{array}{c} R\left(X\right)=\frac{1}{1-\gamma }\log\left\{\frac{B\left(a\gamma -\gamma +1,b\gamma -\gamma +1\right)\;{}_{1}F_{1}\left(a\gamma -\gamma +1;a\gamma +b\gamma -2\gamma +2;-c\gamma \right)}{{\left[B\left(a,b\right)\right]}^{\gamma }{\left[\;{}_{1}F_{1}\left(a;a+b;-c\right)\right]}^{\gamma }}\right\} \end{array}$$
and
$$\begin{array}{c} CE\left(X\right)=-\int_{0}^{1}\left[1-\frac{x^{a}\;\Phi_{1}\left(a,1-b,a+1;x,cx\right)}{aB\left(a,b\right)\;{}_{1}F_{1}\left(a;a+b;-c\right)}\right]\log\left[1-\frac{x^{a}\;\Phi_{1}\left(a,1-b,a+1;x,cx\right)}{aB\left(a,b\right)\;{}_{1}F_{1}\left(a;a+b;-c\right)}\right]dx \end{array}$$
for 0<x<1, a>0, b>0 and c>0.

14. Libby and Novick’s beta distribution [27]: for this distribution,
$$\begin{array}{c} f_{X}\left(x\right)=\frac{c^{a}x^{a-1}{\left(1-x\right)}^{b-1}}{B\left(a,b\right){\left[1-(1-c)x\right]}^{a+b}}, \end{array}$$
$$\begin{array}{c} F_{X}\left(x\right)=I_{\frac{1-x}{1+cx-x}}\left(b,a\right), \end{array}$$
$$\begin{array}{c} GM\left(X\right)=\frac{c^{a}}{B(a,b)}{\left.\frac{\partial }{\partial \alpha }\left[B\left(\alpha +a,b\right)\;{}_{2}F_{1}\left(\alpha +a,a+b;\alpha +a+b;1-c\right)\right]\right|}_{\alpha =0}, \end{array}$$
$$\begin{array}{ccc} & & S\left(X\right)=-a\log c+\frac{\left(1-a\right)c^{a}}{B(a,b)}{\left.\frac{\partial }{\partial \alpha }\left[B\left(\alpha +a,b\right)\;{}_{2}F_{1}\left(\alpha +a,a+b;\alpha +a+b;1-c\right)\right]\right|}_{\alpha =0}\\ & & \;\;+\frac{\left(1-b\right)c^{a}}{B(a,b)}{\left.\frac{\partial }{\partial \alpha }\left[B\left(a,b+\alpha \right)\;{}_{2}F_{1}\left(a,a+b;\alpha +a+b;1-c\right)\right]\right|}_{\alpha =0}\\ & & \;\;+c^{a}\left(a+b\right){\left.\frac{\partial }{\partial \alpha }\;{}_{2}F_{1}\left(a,a+b-\alpha ;a+b;1-c\right)\right|}_{\alpha =0}, \end{array}$$
$$\begin{array}{c} R\left(X\right)=\frac{1}{1-\gamma }\log\left\{\frac{c^{a\gamma }B\left(a\gamma -\gamma +1,b\gamma -\gamma +1\right)}{B{\left[(a,b)\right]}^{\gamma }}\;{}_{2}F_{1}\left(a\gamma -\gamma +1,a+b;a\gamma +b\gamma -2\gamma +2;1-c\right)\right\} \end{array}$$
and
$$\begin{array}{c} CE\left(X\right)=-\int_{0}^{1}I_{\frac{cx}{1+cx-x}}\left(a,b\right)\log I_{\frac{cx}{1+cx-x}}\left(a,b\right)dx \end{array}$$
for 0<x<1, a>0, b>0, and c>0.

15. Generalized beta distribution [28]: for this distribution,
$$\begin{array}{c} f_{X}\left(x\right)=\frac{|a|x^{ap-1}{\left[1-\left(1-c\right){\left(\frac{x}{b}\right)}^{a}\right]}^{q-1}}{b^{ap}B\left(p,q\right){\left[1+c{\left(\frac{x}{b}\right)}^{a}\right]}^{p+q}}, \end{array}$$
$$\begin{array}{c} F_{X}\left(x\right)=\frac{x^{ap}}{pB\left(p,q\right)b^{ap}}F_{1}\left(p,1-q,p+q,p+1;\left(1-c\right){\left(\frac{x}{b}\right)}^{a},-c{\left(\frac{x}{b}\right)}^{a}\right), \end{array}$$
$$\begin{array}{c} GM\left(X\right)={\left.\frac{\partial }{\partial \alpha }\left[\frac{b^{\alpha }B\left(p+\frac{\alpha }{a},q\right)}{B(p,q)}\;{}_{2}F_{1}\left(p+\frac{\alpha }{a},\frac{\alpha }{a};p+q+\frac{\alpha }{a};c\right)\right]\right|}_{\alpha =0}, \end{array}$$
$$\begin{array}{ccc} & & S(X)=-\log|a|+ap\log b+\log B(p,q)\\ & & \;\;+\left(1-ap\right){\left.\frac{\partial }{\partial \alpha }\left[\frac{b^{\alpha }B\left(p+\frac{\alpha }{a},q\right)}{B(p,q)}\;{}_{2}F_{1}\left(p+\frac{\alpha }{a},\frac{\alpha }{a};p+q+\frac{\alpha }{a};c\right)\right]\right|}_{\alpha =0}\\ & & \;\;-\left(1-q\right){\left.\frac{\partial }{\partial \alpha }\left[\frac{B\left(p,q+\alpha \right)}{B(p,q)}\;{}_{2}F_{1}\left(p,\alpha ;p+q+\alpha ;c\right)\right]\right|}_{\alpha =0}\\ & & \;\;+\left(p+q\right){\left.\frac{\partial }{\partial \alpha }\;{}_{2}F_{1}\left(p,\alpha ;p+q;c\right)\right|}_{\alpha =0}, \end{array}$$
$$\begin{array}{ccc} & & R\left(X\right)=\frac{1}{1-\gamma }\log\{\frac{b^{1-\gamma }B\left(p\gamma +\frac{1-\gamma }{a},q\gamma -\gamma +1\right)}{{\left[B(p,q)\right]}^{\gamma }}\\ & & \;\;\;\;\;\cdot {}_{2}F_{1}\left(p\gamma +\frac{1-\gamma }{a},\frac{\left(a+1)(1-\gamma \right)}{a};p\gamma +q\gamma +\left(1-\gamma \right)\left(\frac{1}{a}+1\right);c\right)\} \end{array}$$
and
$$\begin{array}{ccc} & & CE\left(X\right)=-\int_{0}^{1}\left[1-\frac{x^{ap}}{pB\left(p,q\right)b^{ap}}F_{1}\left(p,1-q,p+q,p+1;\left(1-c\right){\left(\frac{x}{b}\right)}^{a},-c{\left(\frac{x}{b}\right)}^{a}\right)\right]\\ & & \;\;\;\;\cdot \log\left[1-\frac{x^{ap}}{pB\left(p,q\right)b^{ap}}F_{1}\left(p,1-q,p+q,p+1;\left(1-c\right){\left(\frac{x}{b}\right)}^{a},-c{\left(\frac{x}{b}\right)}^{a}\right)\right]dx \end{array}$$
for $0<x^{a}<\frac{b^{a}}{1-c}$, b>0, 0<c<1, p>0 and q>0.

16. Log-logistic distribution: for this distribution,
$$\begin{array}{c} f_{X}\left(x\right)=\frac{ba^{b}x^{b-1}}{{\left(a^{b}+x^{b}\right)}^{2}}, \end{array}$$
$$\begin{array}{c} F_{X}\left(x\right)=\frac{x^{b}}{a^{b}+x^{b}}, \end{array}$$
$$\begin{array}{c} GM(X)=\log a, \end{array}$$
$$\begin{array}{c} S(X)=\log a-\log b+2, \end{array}$$
$$\begin{array}{c} R\left(X\right)=\log a-\log b+\frac{2b\gamma \log a}{1-\gamma }+\frac{1}{1-\gamma }\log B\left(\gamma +\frac{\gamma -1}{b},\gamma +\frac{1-\gamma }{b}\right) \end{array}$$
and
$$\begin{array}{c} CE\left(X\right)=-\frac{a}{b}\Gamma \left(\frac{1}{b}\right)\left[\Gamma^{{}^{\prime }}\left(1-\frac{1}{b}\right)-\Gamma^{{}^{\prime }}\left(1\right)\Gamma \left(1-\frac{1}{b}\right)\right] \end{array}$$
for x>0, a>0 and b>0.

17. Inverse Gaussian distribution [29]: for this distribution,
$$\begin{array}{c} f_{X}\left(x\right)=\sqrt{\frac{a}{2\pi x^{3}}}\exp\left[-\frac{a{\left(x-b\right)}^{2}}{2b^{2}x}\right], \end{array}$$
$$\begin{array}{c} F_{X}\left(x\right)=\Phi \left(\sqrt{\frac{a}{x}}\left(\frac{x}{b}-1\right)\right)-\Phi \left(-\sqrt{\frac{a}{x}}\left(\frac{x}{b}+1\right)\right)\exp\left(\frac{2a}{b}\right), \end{array}$$
$$\begin{array}{c} GM\left(X\right)=\sqrt{\frac{2a\log b}{\pi b}}\exp\left(\frac{a}{b}\right)K_{-\frac{1}{2}}\exp\left(\frac{a}{b}\right)+\sqrt{\frac{2a}{\pi b}}\exp\left(\frac{a}{b}\right){\left.\frac{\partial }{\partial \alpha }K_{\alpha -\frac{1}{2}}\exp\left(\frac{a}{b}\right)\right|}_{\alpha =0}, \end{array}$$
$$\begin{array}{ccc} & & S\left(X\right)=\frac{1}{2}-\frac{1}{2}\log\left(\frac{a}{2\pi }\right)+3\sqrt{\frac{a\log b}{2\pi b}}\exp\left(\frac{a}{b}\right)K_{-\frac{1}{2}}\exp\left(\frac{a}{b}\right)\\ & & \;\;+3\sqrt{\frac{a}{2\pi b}}\exp\left(\frac{a}{b}\right){\left.\frac{\partial }{\partial \alpha }K_{\alpha -\frac{1}{2}}\exp\left(\frac{a}{b}\right)\right|}_{\alpha =0}, \end{array}$$
$$\begin{array}{c} R\left(X\right)=\frac{\gamma }{2(1-\gamma )}\log\frac{a}{2\pi b^{3}}+\frac{1}{1-\gamma }\log\frac{2b}{\sqrt{\gamma }}+\frac{a}{(1-\gamma )b}+\frac{1}{1-\gamma }K_{1-\frac{3\gamma }{2}}\left(\frac{a}{b}\right) \end{array}$$
and
$$\begin{array}{ccc} & & CE\left(X\right)=-\int_{0}^{\infty }\left[\Phi \left(\sqrt{\frac{a}{x}}\left(1-\frac{x}{b}\right)\right)-\Phi \left(-\sqrt{\frac{a}{x}}\left(\frac{x}{b}+1\right)\right)\exp\left(\frac{2a}{b}\right)\right]\\ & & \;\;\;\;\cdot \log\left[\Phi \left(\sqrt{\frac{a}{x}}\left(1-\frac{x}{b}\right)\right)-\Phi \left(-\sqrt{\frac{a}{x}}\left(\frac{x}{b}+1\right)\right)\exp\left(\frac{2a}{b}\right)\right]dx \end{array}$$
for x>0, a>0 and b>0.

18. Gompertz distribution [30]: for this distribution,
$$\begin{array}{c} f_{X}\left(x\right)=ab\exp\left[a+bx-a\exp(bx)\right], \end{array}$$
$$\begin{array}{c} F_{X}\left(x\right)=1-\exp\left[a-a\exp(bx)\right], \end{array}$$
$$\begin{array}{c} GM\left(X\right)=-\log b+a\exp\left(a\right)\int_{1}^{\infty }\log\log y\exp\left(-ay\right)dy, \end{array}$$
$$\begin{array}{c} S\left(X\right)=-a-\log\left(ab\right)-\mathrm{Ei}\left(-a\right)\exp\left(a\right)+a^{2}\exp\left(a\right)\int_{1}^{\infty }t\exp\left(at\right)dt, \end{array}$$
$$\begin{array}{c} R\left(X\right)=-\log b-\frac{\gamma \log\gamma }{1-\gamma }+\frac{a\gamma }{1-\gamma }+\frac{\log\Gamma (\gamma ,a\gamma )}{1-\gamma } \end{array}$$
and
$$\begin{array}{c} CE\left(X\right)=\frac{1-a\exp(a)\mathrm{Ei}(-a)}{b} \end{array}$$
for x>0, a>0 and b>0.

19. Exponential distribution: for this distribution,
$$\begin{array}{c} f_{X}\left(x\right)=a\exp\left(-ax\right), \end{array}$$
$$\begin{array}{c} F_{X}\left(x\right)=1-\exp\left(-ax\right), \end{array}$$
$$\begin{array}{c} GM\left(X\right)=\Gamma^{{}^{\prime }}\left(1\right)-\log a, \end{array}$$
$$\begin{array}{c} S(X)=1-\log a, \end{array}$$
$$\begin{array}{c} R\left(X\right)=-\log a-\frac{\log\gamma }{1-\gamma } \end{array}$$
and
$$\begin{array}{c} CE\left(X\right)=\frac{1}{a} \end{array}$$
for x>0 and a>0.

20. Inverse exponential distribution: for this distribution,
$$\begin{array}{c} f_{X}\left(x\right)=bx^{-2}\exp\left(-\frac{b}{x}\right), \end{array}$$
$$\begin{array}{c} F_{X}\left(x\right)=\exp\left(-\frac{b}{x}\right), \end{array}$$
$$\begin{array}{c} GM\left(X\right)=\log b-\Gamma^{{}^{\prime }}\left(1\right), \end{array}$$
$$\begin{array}{c} S\left(X\right)=\log b-2\Gamma^{{}^{\prime }}\left(1\right)+1, \end{array}$$
$$\begin{array}{c} R\left(X\right)=\log b+\frac{1-2\gamma }{1-\gamma }\log\gamma +\frac{\log\Gamma (2\gamma -1)}{1-\gamma } \end{array}$$
and
$$\begin{array}{c} CE\left(X\right)=-\int_{0}^{\infty }\left[1-\exp\left(-\frac{b}{x}\right)\right]\log\left[1-\exp\left(-\frac{b}{x}\right)\right]dx \end{array}$$
for x>0 and b>0.

21. Exponentiated exponential distribution [31]: for this distribution,
$$\begin{array}{c} f_{X}\left(x\right)=ab\exp\left(-bx\right){\left[1-\exp(-bx)\right]}^{a-1}, \end{array}$$
$$\begin{array}{c} F_{X}\left(x\right)={\left[1-\exp(-bx)\right]}^{a}, \end{array}$$
$$\begin{array}{c} GM\left(X\right)=a\int_{0}^{1}{\left(1-y\right)}^{a-1}\log\left(-\log y\right)dy-\log b, \end{array}$$
$$\begin{array}{c} S\left(X\right)=-\log\left(ab\right)-\Gamma^{{}^{\prime }}\left(1\right)+\frac{\Gamma^{{}^{\prime }}\left(a+1\right)}{\Gamma (a+1)}+\frac{a-1}{a}, \end{array}$$
$$\begin{array}{c} R\left(X\right)=\log b+\frac{\gamma \log a}{1-\gamma }+\frac{1}{1-\gamma }\log B\left(\gamma ,a\gamma -\gamma +1\right) \end{array}$$
and
$$\begin{array}{c} CE\left(X\right)=\frac{1}{b}\sum_{k=1}^{\infty }\frac{1}{k}B\left(0,ak+1\right)-\frac{1}{b}\sum_{k=1}^{\infty }\frac{1}{k}B\left(0,ak+a+1\right) \end{array}$$
for x>0 and a>0.

22. Gamma distribution: for this distribution,
$$\begin{array}{c} f_{X}\left(x\right)=\frac{b^{a}x^{a-1}\exp\left(-bx\right)}{\Gamma (a)}, \end{array}$$
$$\begin{array}{c} F_{X}\left(x\right)=\frac{\gamma \left(a,bx\right)}{\Gamma (a)}, \end{array}$$
$$\begin{array}{c} GM\left(X\right)=\frac{\Gamma^{{}^{\prime }}\left(a\right)-\Gamma \left(a\right)\log b}{{\left[\Gamma \left(a\right)\right]}^{2}}, \end{array}$$
$$\begin{array}{c} S\left(X\right)=a-a\log b+\log\Gamma \left(a\right)+\left(1-a\right)\left\{\frac{\Gamma^{{}^{\prime }}\left(a\right)}{{\left[\Gamma (a)\right]}^{2}}-\frac{\log b}{\Gamma (a)}\right\}, \end{array}$$
$$\begin{array}{c} R\left(X\right)=\log b+\frac{\gamma -a\gamma -1}{1-\gamma }\log\gamma +\frac{1}{1-\gamma }\log\Gamma \left(a\gamma -\gamma +1\right)-\frac{\gamma }{1-\gamma }\log\Gamma \left(a\right) \end{array}$$
and
$$\begin{array}{c} CE\left(X\right)=-\int_{0}^{\infty }\frac{\Gamma \left(a,bx\right)}{\Gamma (a)}\log\frac{\Gamma \left(a,bx\right)}{\Gamma (a)}dx \end{array}$$
for x>0, a>0 and b>0.

23. Chisquare distribution: for this distribution,
$$\begin{array}{c} f_{X}\left(x\right)=\frac{x^{\frac{k}{2}-1}\exp\left(-\frac{x}{2}\right)}{2^{\frac{k}{2}}\Gamma \left(\frac{k}{2}\right)}, \end{array}$$
$$\begin{array}{c} F_{X}\left(x\right)=\frac{\gamma \left(\frac{k}{2},\frac{x}{2}\right)}{\Gamma \left(\frac{k}{2}\right)}, \end{array}$$
$$\begin{array}{c} GM\left(X\right)=\frac{\Gamma^{{}^{\prime }}\left(\frac{k}{2}\right)+\Gamma \left(\frac{k}{2}\right)\log2}{{\left[\Gamma \left(\frac{k}{2}\right)\right]}^{2}}, \end{array}$$
$$\begin{array}{c} S\left(X\right)=\frac{k}{2}+\frac{k}{2}\log2+\log\Gamma \left(\frac{k}{2}\right)+\frac{k}{2}\left\{\frac{\Gamma^{{}^{\prime }}\left(\frac{k}{2}\right)}{{\left[\Gamma \left(\frac{k}{2}\right)\right]}^{2}}+\frac{\log2}{\Gamma \left(\frac{k}{2}\right)}\right\}, \end{array}$$
$$\begin{array}{c} R\left(X\right)=-\log2+\frac{\gamma -\frac{k}{2}\gamma -1}{1-\gamma }\log\gamma +\frac{1}{1-\gamma }\log\Gamma \left(\frac{k}{2}\gamma -\gamma +1\right)-\frac{\gamma }{1-\gamma }\log\Gamma \left(\frac{k}{2}\right) \end{array}$$
and
$$\begin{array}{c} CE\left(X\right)=-\int_{0}^{\infty }\frac{\Gamma \left(\frac{k}{2},\frac{x}{2}\right)}{\Gamma \left(\frac{k}{2}\right)}\log\frac{\Gamma \left(\frac{k}{2},\frac{x}{2}\right)}{\Gamma \left(\frac{k}{2}\right)}dx \end{array}$$
for x>0 and k>0.

24. Chi distribution: for this distribution,
$$\begin{array}{c} f_{X}\left(x\right)=\frac{x^{k-1}\exp\left(-\frac{x^{2}}{2}\right)}{2^{\frac{k}{2}-1}\Gamma \left(\frac{k}{2}\right)}, \end{array}$$
$$\begin{array}{c} F_{X}\left(x\right)=\frac{\gamma \left(\frac{k}{2},\frac{x^{2}}{2}\right)}{\Gamma \left(\frac{k}{2}\right)}, \end{array}$$
$$\begin{array}{c} GM\left(X\right)=\frac{\Gamma \left(\frac{k}{2}\right)\log2+\Gamma^{{}^{\prime }}\left(\frac{k}{2}\right)}{2\Gamma \left(\frac{k}{2}\right)}, \end{array}$$
$$\begin{array}{c} S\left(X\right)=-\frac{1}{2}\log2+\frac{k}{2}-\frac{1-k}{2}\frac{\Gamma^{{}^{\prime }}\left(\frac{k}{2}\right)}{\Gamma \left(\frac{k}{2}\right)}+\log\Gamma \left(\frac{k}{2}\right), \end{array}$$
$$\begin{array}{c} R\left(X\right)=\frac{1}{1-\gamma }\log\left\{\frac{2^{\frac{\gamma -1}{2}}\gamma^{\frac{\gamma -\gamma k-1}{2}}}{{\left[\Gamma \left(\frac{k}{2}\right)\right]}^{\gamma }}\Gamma \left(\frac{\gamma k-\gamma +1}{2}\right)\right\} \end{array}$$
and
$$\begin{array}{c} CE\left(X\right)=-\int_{0}^{\infty }\frac{\Gamma \left(\frac{k}{2},\frac{x^{2}}{2}\right)}{\Gamma \left(\frac{k}{2}\right)}\log\frac{\Gamma \left(\frac{k}{2},\frac{x^{2}}{2}\right)}{\Gamma \left(\frac{k}{2}\right)}dx \end{array}$$
for x>0 and k>0.

25. Inverse gamma distribution: for this distribution,
$$\begin{array}{c} f_{X}\left(x\right)=\frac{b^{a}x^{-a-1}\exp\left(-\frac{b}{x}\right)}{\Gamma (a)}, \end{array}$$
$$\begin{array}{c} F_{X}\left(x\right)=\frac{\Gamma \left(a,\frac{b}{x}\right)}{\Gamma (a)}, \end{array}$$
$$\begin{array}{c} GM\left(X\right)=\frac{\Gamma \left(a\right)\log b-\Gamma^{{}^{\prime }}\left(a\right)}{\Gamma \left(a\right)}, \end{array}$$
$$\begin{array}{c} S\left(X\right)=-a\log b+\left(a+1\right)\frac{\log b-\Gamma^{{}^{\prime }}\left(1\right)}{\Gamma (a)}+a+\log\Gamma \left(a\right), \end{array}$$
$$\begin{array}{c} R\left(X\right)=\log b+\frac{1-\gamma -a\gamma }{1-\gamma }\log\gamma -\frac{\gamma }{1-\gamma }\log\Gamma \left(a\right)+\frac{\log\Gamma (a\gamma +\gamma -1)}{1-\gamma } \end{array}$$
and
$$\begin{array}{c} CE\left(X\right)=-\int_{0}^{\infty }\frac{\gamma \left(a,\frac{b}{x}\right)}{\Gamma (a)}\log\frac{\gamma \left(a,\frac{b}{x}\right)}{\Gamma (a)}dx \end{array}$$
for x>0, a>0 and b>0.

26. Inverse chisquare distribution: for this distribution,
$$\begin{array}{c} f_{X}\left(x\right)=\frac{x^{-\frac{k}{2}-1}\exp\left(-\frac{1}{2x}\right)}{2^{\frac{k}{2}}\Gamma \left(\frac{k}{2}\right)}, \end{array}$$
$$\begin{array}{c} F_{X}\left(x\right)=\frac{\Gamma \left(\frac{k}{2},\frac{1}{2x}\right)}{\Gamma \left(\frac{k}{2}\right)}, \end{array}$$
$$\begin{array}{c} GM\left(X\right)=\frac{-\Gamma \left(\frac{k}{2}\right)\log2-\Gamma^{{}^{\prime }}\left(\frac{k}{2}\right)}{\Gamma \left(\frac{k}{2}\right)}, \end{array}$$
$$\begin{array}{c} S\left(X\right)=\frac{k}{2}\log2-\left(\frac{k}{2}+1\right)\frac{\log2+\Gamma^{{}^{\prime }}\left(1\right)}{\Gamma \left(\frac{k}{2}\right)}+\frac{k}{2}+\log\Gamma \left(\frac{k}{2}\right), \end{array}$$
$$\begin{array}{c} R\left(X\right)=-\log2+\frac{1-\gamma -\frac{k\gamma }{2}}{1-\gamma }\log\gamma -\frac{\gamma }{1-\gamma }\log\Gamma \left(\frac{k}{2}\right)+\frac{\log\Gamma \left(\frac{k}{2}\gamma +\gamma -1\right)}{1-\gamma } \end{array}$$
and
$$\begin{array}{c} CE\left(X\right)=-\int_{0}^{\infty }\frac{\gamma \left(\frac{k}{2},\frac{1}{2x}\right)}{\Gamma \left(\frac{k}{2}\right)}\log\frac{\gamma \left(\frac{k}{2},\frac{1}{2x}\right)}{\Gamma \left(\frac{k}{2}\right)}dx \end{array}$$
for x>0 and k>0.

27. Inverse chi distribution: for this distribution,
$$\begin{array}{c} f_{X}\left(x\right)=\frac{x^{-k-1}\exp\left(-\frac{1}{2x^{2}}\right)}{2^{\frac{k}{2}-1}\Gamma \left(\frac{k}{2}\right)}, \end{array}$$
$$\begin{array}{c} F_{X}\left(x\right)=\frac{\Gamma \left(\frac{k}{2},\frac{1}{2x^{2}}\right)}{\Gamma \left(\frac{k}{2}\right)}, \end{array}$$
$$\begin{array}{c} GM\left(X\right)=-\frac{\Gamma \left(\frac{k}{2}\right)\log2+\Gamma^{{}^{\prime }}\left(\frac{k}{2}\right)}{2\Gamma \left(\frac{k}{2}\right)}, \end{array}$$
$$\begin{array}{c} S\left(X\right)=-\frac{3}{2}\log2-\frac{k}{2}-\frac{1+k}{2}\frac{\Gamma^{{}^{\prime }}\left(\frac{k}{2}\right)}{\Gamma \left(\frac{k}{2}\right)}+\log\Gamma \left(\frac{k}{2}\right), \end{array}$$
$$\begin{array}{c} R\left(X\right)=\frac{1}{1-\gamma }\log\left\{\frac{2^{\frac{3(\gamma -1)}{2}}\gamma^{\frac{1-\gamma -\gamma k}{2}}}{{\left[\Gamma \left(\frac{k}{2}\right)\right]}^{\gamma }}\Gamma \left(\frac{\gamma +\gamma k-1}{2}\right)\right\} \end{array}$$
and
$$\begin{array}{c} CE\left(X\right)=-\int_{0}^{\infty }\frac{\gamma \left(\frac{k}{2},\frac{1}{2x^{2}}\right)}{\Gamma \left(\frac{k}{2}\right)}\log\frac{\gamma \left(\frac{k}{2},\frac{1}{2x^{2}}\right)}{\Gamma \left(\frac{k}{2}\right)}dx \end{array}$$
for x>0 and k>0.

28. Rayleigh distribution: for this distribution,
$$\begin{array}{c} f_{X}\left(x\right)=2b^{2}x\exp\left[-{\left(bx\right)}^{2}\right], \end{array}$$
$$\begin{array}{c} F_{X}\left(x\right)=1-\exp\left[-{\left(bx\right)}^{2}\right], \end{array}$$
$$\begin{array}{c} GM\left(X\right)=\frac{1}{2}\Gamma^{{}^{\prime }}\left(1\right)-\log b, \end{array}$$
$$\begin{array}{c} S\left(X\right)=1-\log\left(2b\right)-\frac{1}{2}\Gamma^{{}^{\prime }}\left(1\right), \end{array}$$
$$\begin{array}{c} R\left(X\right)=-\log\left(2b\right)-\frac{\log\gamma }{2}-\frac{\gamma \log\gamma }{1-\gamma }+\frac{1}{1-\gamma }\log\left(\frac{1+\gamma }{2}\right) \end{array}$$
and
$$\begin{array}{c} CE\left(X\right)=\frac{\sqrt{\pi }}{4b} \end{array}$$
for x>0 and b>0.

29. Weibull distribution [32]: for this distribution,
$$\begin{array}{c} f_{X}\left(x\right)=ab^{a}x^{a-1}\exp\left[-{\left(bx\right)}^{a}\right], \end{array}$$
$$\begin{array}{c} F_{X}\left(x\right)=1-\exp\left[-{\left(bx\right)}^{a}\right], \end{array}$$
$$\begin{array}{c} GM\left(X\right)=\frac{1}{a}\Gamma^{{}^{\prime }}\left(1\right)-\log b, \end{array}$$
$$\begin{array}{c} S\left(X\right)=1-\log\left(ab\right)+\frac{1-a}{a}\Gamma^{{}^{\prime }}\left(1\right), \end{array}$$
$$\begin{array}{c} R\left(X\right)=-\log\left(ab\right)-\frac{\log\gamma }{a}-\frac{\gamma \log\gamma }{1-\gamma }+\frac{1}{1-\gamma }\log\left(\frac{1-\gamma }{a}+\gamma \right) \end{array}$$
and
$$\begin{array}{c} CE\left(X\right)=\frac{1}{ab}\Gamma \left(1+\frac{1}{a}\right) \end{array}$$
for x>0, a>0 and b>0.

30. Inverse Rayleigh distribution: for this distribution,
$$\begin{array}{c} f_{X}\left(x\right)=2ax^{-3}\exp\left(-ax^{-2}\right), \end{array}$$
$$\begin{array}{c} F_{X}\left(x\right)=\exp\left(-ax^{-2}\right), \end{array}$$
$$\begin{array}{c} GM\left(X\right)=\frac{\log a-\Gamma^{{}^{\prime }}\left(1\right)}{2}, \end{array}$$
$$\begin{array}{c} S\left(X\right)=1+\frac{\log a}{2}-\log2-\frac{\Gamma^{{}^{\prime }}\left(1\right)}{b}, \end{array}$$
$$\begin{array}{c} R\left(X\right)=\frac{1}{2}\log a-\log2+\frac{3\gamma -1}{2}\frac{\log\gamma }{1-\gamma }+\frac{1}{1-\gamma }\log\Gamma \left(\frac{3\gamma -1}{2}\right) \end{array}$$
and
$$\begin{array}{c} CE\left(X\right)=\frac{\sqrt{a}}{2}\Gamma \left(-\frac{1}{2}\right)\sum_{k=1}^{\infty }k^{-\frac{1}{2}}-\frac{\sqrt{a}}{2}\Gamma \left(-\frac{1}{2}\right)\sum_{k=1}^{\infty }\frac{\sqrt{k+1}}{k} \end{array}$$
for x>0 and a>0.

31. Inverse Weibull distribution: for this distribution,
$$\begin{array}{c} f_{X}\left(x\right)=abx^{-b-1}\exp\left(-ax^{-b}\right), \end{array}$$
$$\begin{array}{c} F_{X}\left(x\right)=\exp\left(-ax^{-b}\right), \end{array}$$
$$\begin{array}{c} GM\left(X\right)=\frac{\log a-\Gamma^{{}^{\prime }}\left(1\right)}{b}, \end{array}$$
$$\begin{array}{c} S\left(X\right)=1+\frac{\log a}{b}-\log b-\frac{\Gamma^{{}^{\prime }}\left(1\right)}{b}, \end{array}$$
$$\begin{array}{c} R\left(X\right)=\frac{1}{b}\log a-\log b+\left(\gamma +\frac{\gamma -1}{b}\right)\frac{\log\gamma }{1-\gamma }+\frac{1}{1-\gamma }\log\Gamma \left(\gamma +\frac{\gamma -1}{b}\right) \end{array}$$
and
$$\begin{array}{c} CE\left(X\right)=\frac{a^{\frac{1}{b}}}{b}\Gamma \left(-\frac{1}{b}\right)\sum_{k=1}^{\infty }k^{\frac{1}{b}-1}-\frac{a^{\frac{1}{b}}}{b}\Gamma \left(-\frac{1}{b}\right)\sum_{k=1}^{\infty }\frac{{\left(k+1\right)}^{\frac{1}{b}}}{k} \end{array}$$
for x>0, a>0 and b>0.

32. Gumbel distribution [33]: for this distribution,
$$\begin{array}{c} f_{X}\left(x\right)=\frac{1}{a}\exp\left(-\frac{x-b}{a}\right)\exp\left[-\exp\left(-\frac{x-b}{a}\right)\right], \end{array}$$
$$\begin{array}{c} F_{X}\left(x\right)=\exp\left[-\exp\left(-\frac{x-b}{a}\right)\right], \end{array}$$
$$\begin{array}{c} GM\left(X\right)=\frac{1}{a}\int_{-\infty }^{\infty }\log x\exp\left(-\frac{x-b}{a}\right)\exp\left[-\exp\left(-\frac{x-b}{a}\right)\right]dx, \end{array}$$
$$\begin{array}{c} S\left(X\right)=1+\log a-\Gamma^{{}^{\prime }}\left(1\right), \end{array}$$
$$\begin{array}{c} R\left(X\right)=\frac{1}{1-\gamma }\log\left[\frac{\Gamma (\gamma )}{a^{\gamma -1}\gamma^{\gamma }}\right] \end{array}$$
and
$$\begin{array}{c} CE\left(X\right)=\int_{-\infty }^{\infty }\left\{1-\exp\left[-\exp\left(-\frac{x-b}{a}\right)\right]\right\}\log\left\{1-\exp\left[-\exp\left(-\frac{x-b}{a}\right)\right]\right\}dx \end{array}$$
for −∞<x<∞, a>0 and −∞<b<∞.

33. Generalized extreme value distribution [34]: for this distribution,
$$\begin{array}{c} f_{X}\left(x\right)=\frac{1}{a}{\left(1+\xi \frac{x-b}{a}\right)}^{-\frac{\xi +1}{\xi }}\exp\left[-{\left(1+\xi \frac{x-b}{a}\right)}^{-\frac{1}{\xi }}\right], \end{array}$$
$$\begin{array}{c} F_{X}\left(x\right)=\exp\left[-{\left(1+\xi \frac{x-b}{a}\right)}^{-\frac{1}{\xi }}\right], \end{array}$$
$$\begin{array}{c} GM\left(X\right)=\frac{1}{a}\int_{-\infty }^{\infty }\log x{\left(1+\xi \frac{x-b}{a}\right)}^{-\frac{\xi +1}{\xi }}\exp\left[-{\left(1+\xi \frac{x-b}{a}\right)}^{-\frac{1}{\xi }}\right]dx, \end{array}$$
$$\begin{array}{c} S\left(X\right)=1+\log a-\left(\xi +1\right)\Gamma^{{}^{\prime }}\left(1\right), \end{array}$$
$$\begin{array}{c} R\left(X\right)=\frac{1}{1-\gamma }\log\left[\frac{\Gamma \left(\gamma \xi -\xi +\gamma \right)}{a^{\gamma -1}\gamma^{\gamma \xi -\xi +\gamma }}\right] \end{array}$$
and
$$\begin{array}{c} CE\left(X\right)=a\Gamma \left(-\xi \right)\sum_{k=1}^{\infty }k^{\xi }-a\Gamma \left(-\xi \right)\sum_{k=1}^{\infty }\frac{{\left(k+1\right)}^{\xi +1}}{k} \end{array}$$
for $b-\frac{a}{\xi }<x<\infty$ if ξ>0, $-\infty <x<b-\frac{a}{\xi }$ if ξ<0, −∞<b<∞ and a>0.

34. Generalized gamma distribution [35]: for this distribution,
$$\begin{array}{c} f_{X}\left(x\right)=\frac{pa^{d}x^{d-1}\exp\left[-{\left(ax\right)}^{p}\right]}{\Gamma \left(\frac{d}{p}\right)}, \end{array}$$
$$\begin{array}{c} F_{X}\left(x\right)=\frac{\gamma \left(\frac{d}{p},{\left(ax\right)}^{p}\right)}{\Gamma \left(\frac{d}{p}\right)}, \end{array}$$
$$\begin{array}{c} GM\left(X\right)=\frac{\frac{1}{p}\Gamma^{{}^{\prime }}\left(\frac{d}{p}\right)-\Gamma \left(\frac{d}{p}\right)\log a}{{\left[\Gamma \left(\frac{d}{p}\right)\right]}^{3}}, \end{array}$$
$$\begin{array}{c} S\left(X\right)=-\log p-d\log a+\Gamma \left(1+\frac{d}{p}\right)+\log\Gamma \left(\frac{d}{p}\right)+\frac{1-d}{p}\frac{\Gamma^{{}^{\prime }}\left(\frac{d}{p}\right)}{{\left[\Gamma \left(\frac{d}{p}\right)\right]}^{3}}-\frac{(1-d)\log a}{{\left[\Gamma \left(\frac{d}{p}\right)\right]}^{2}}, \end{array}$$
$$\begin{array}{c} R\left(X\right)=-\frac{\log p}{1-\gamma }-\log a-\frac{d\gamma -\gamma +1}{p}\frac{\log\gamma }{1-\gamma }-\frac{\gamma }{1-\gamma }\log\Gamma \left(\frac{d}{p}\right)+\frac{1}{1-\gamma }\log\Gamma \left(\frac{d\gamma -\gamma +1}{p}\right) \end{array}$$
and
$$\begin{array}{c} CE\left(X\right)=-\int_{0}^{\infty }\frac{\Gamma \left(\frac{d}{p},{\left(ax\right)}^{p}\right)}{\Gamma \left(\frac{d}{p}\right)}\log\frac{\Gamma \left(\frac{d}{p},{\left(ax\right)}^{p}\right)}{\Gamma \left(\frac{d}{p}\right)}dx \end{array}$$
for x>0, a>0, d>0 and p>0.

35. Pareto distribution of type I [36]: for this distribution,
$$\begin{array}{c} f_{X}\left(x\right)=\frac{aK^{a}}{x^{a+1}}, \end{array}$$
$$\begin{array}{c} F_{X}\left(x\right)=1-{\left(\frac{K}{x}\right)}^{a}, \end{array}$$
$$\begin{array}{c} GM\left(X\right)=\log K+\frac{1}{a}, \end{array}$$
$$\begin{array}{c} S\left(X\right)=1-a+\frac{1}{a}+\log K, \end{array}$$
$$\begin{array}{c} R\left(X\right)=\log K+\frac{\gamma \log a}{1-\gamma }-\frac{\log\left(a\gamma +\gamma -1\right)}{1-\gamma } \end{array}$$
and
$$\begin{array}{c} CE\left(X\right)=-\frac{Ka}{{\left(a-1\right)}^{2}} \end{array}$$
for x≥K, K>0 and a>0.

36. Pareto distribution of type II [37]: for this distribution,
$$\begin{array}{c} f_{X}\left(x\right)=\frac{ab^{a}}{{\left(x+b\right)}^{a+1}}, \end{array}$$
$$\begin{array}{c} F_{X}\left(x\right)=1-\frac{b^{a}}{{\left(x+b\right)}^{a}}, \end{array}$$
$$\begin{array}{c} GM\left(X\right)=\log b+\Gamma^{{}^{\prime }}\left(1\right)-\frac{\Gamma^{{}^{\prime }}\left(a\right)}{\Gamma (a)}, \end{array}$$
$$\begin{array}{c} S(X)=-\log a-a\log b+(a+1)(a\log b+1), \end{array}$$
$$\begin{array}{c} R\left(X\right)=\log b+\frac{\gamma \log a}{1-\gamma }-\frac{\log(a\gamma +\gamma -1)}{1-\gamma } \end{array}$$
and
$$\begin{array}{c} CE\left(X\right)=\frac{ab}{{\left(a-1\right)}^{2}} \end{array}$$
for x>0, a>0 and b>0.

37. Generalized Pareto distribution [38]: for this distribution,
$$\begin{array}{c} f_{X}\left(x\right)={\left(1+\xi x\right)}^{-\frac{\xi +1}{\xi }}, \end{array}$$
$$\begin{array}{c} F_{X}\left(x\right)=1-{\left(1+\xi x\right)}^{-\frac{1}{\xi }}, \end{array}$$
$$\begin{array}{c} GM\left(X\right)=\left\{\begin{array}{cc} -\log\xi +\Gamma^{{}^{\prime }}\left(1\right)-\frac{\Gamma^{{}^{\prime }}\left(\frac{1}{\xi }\right)}{\xi \Gamma \left(1+\frac{1}{\xi }\right)}, & \mathrm{if}\;\xi >0,\\ -\log\left(-\xi \right)+\Gamma^{{}^{\prime }}\left(1\right)-\frac{\Gamma^{{}^{\prime }}\left(1-\frac{1}{\xi }\right)}{\xi \Gamma \left(1-\frac{1}{\xi }\right)}, & \mathrm{if}\;\xi <0, \end{array}\right. \end{array}$$
$$\begin{array}{c} S(X)=\xi +1, \end{array}$$
$$\begin{array}{c} R\left(X\right)=\frac{1}{\gamma (\xi +1)-\xi } \end{array}$$
and
$$\begin{array}{c} CE\left(X\right)=\frac{1}{{\left(\xi -1\right)}^{2}} \end{array}$$
for 0<x<∞ if ξ>0 and $0<x<-\frac{1}{\xi }$ if ξ<0.

38. Uniform distribution: for this distribution,
$$\begin{array}{c} f_{X}\left(x\right)=\frac{1}{b-a}, \end{array}$$
$$\begin{array}{c} F_{X}\left(x\right)=\frac{x-a}{b-a}, \end{array}$$
$$\begin{array}{c} GM\left(X\right)=\frac{b\log b-a\log a-b+a}{b-a}, \end{array}$$
$$\begin{array}{c} S(X)=\log(b-a), \end{array}$$
$$\begin{array}{c} R(X)=\log(b-a) \end{array}$$
and
$$\begin{array}{c} CE\left(X\right)=-\frac{b-a}{4} \end{array}$$
for a<x<b and ∞>b>a>−∞.

39. Power function distribution of type I: for this distribution,
$$\begin{array}{c} f_{X}\left(x\right)=ax^{a-1}, \end{array}$$
$$\begin{array}{c} F_{X}\left(x\right)=x^{a}, \end{array}$$
$$\begin{array}{c} GM\left(X\right)=-\frac{1}{a}, \end{array}$$
$$\begin{array}{c} S\left(X\right)=1-\frac{1}{a}-\log a, \end{array}$$
$$\begin{array}{c} R\left(X\right)=\frac{\gamma \log a}{1-\gamma }-\frac{\log\left(a\gamma -\gamma +1\right)}{1-\gamma } \end{array}$$
and
$$\begin{array}{c} CE\left(X\right)=\psi \left(\frac{1}{a}+2\right)-\psi \left(2\right) \end{array}$$
for 0<x<1 and a>0.

40. Power function distribution of type II: for this distribution,
$$\begin{array}{c} f_{X}\left(x\right)=a{\left(1-x\right)}^{a-1}, \end{array}$$
$$\begin{array}{c} F_{X}\left(x\right)=1-{\left(1-x\right)}^{a}, \end{array}$$
$$\begin{array}{c} GM\left(X\right)=-\frac{1}{a}, \end{array}$$
$$\begin{array}{c} S\left(X\right)=1-\frac{1}{a}-\log a, \end{array}$$
$$\begin{array}{c} R\left(X\right)=\frac{\gamma \log a}{1-\gamma }-\frac{\log\left(a\gamma -\gamma +1\right)}{1-\gamma } \end{array}$$
and
$$\begin{array}{c} CE\left(X\right)=\frac{a}{{\left(a+1\right)}^{2}} \end{array}$$
for 0<x<1 and a>0.

41. Arcsine distribution: for this distribution,
$$\begin{array}{c} f_{X}\left(x\right)=\frac{1}{\pi \sqrt{x(1-x)}}, \end{array}$$
$$\begin{array}{c} F_{X}\left(x\right)=\frac{2}{\pi }\arcsin\left(\sqrt{x}\right), \end{array}$$
$$\begin{array}{c} GM\left(X\right)=\frac{\Gamma^{{}^{\prime }}\left(\frac{1}{2}\right)}{\sqrt{\pi }}-\Gamma^{{}^{\prime }}\left(1\right), \end{array}$$
$$\begin{array}{c} S\left(X\right)=\log\pi +\frac{\Gamma^{{}^{\prime }}\left(\frac{1}{2}\right)}{\sqrt{\pi }}-\Gamma^{{}^{\prime }}\left(1\right), \end{array}$$
$$\begin{array}{c} R\left(X\right)=\frac{1}{1-\gamma }\log\frac{B\left(\frac{\gamma }{2}-\gamma +1,\frac{\gamma }{2}-\gamma +1\right)}{\pi^{\gamma }} \end{array}$$
and
$$\begin{array}{c} CE\left(X\right)=-\int_{0}^{\infty }\left[1-\frac{2}{\pi }\arcsin\left(\sqrt{x}\right)\right]\log\left[1-\frac{2}{\pi }\arcsin\left(\sqrt{x}\right)\right]dx \end{array}$$
for 0<x<1.

42. Beta distribution: for this distribution,
$$\begin{array}{c} f_{X}\left(x\right)=\frac{x^{a-1}{\left(1-x\right)}^{b-1}}{B(a,b)}, \end{array}$$
$$\begin{array}{c} F_{X}\left(x\right)=I_{x}\left(a,b\right), \end{array}$$
$$\begin{array}{c} GM\left(X\right)=\frac{\Gamma^{{}^{\prime }}\left(a\right)}{\Gamma \left(a\right)}-\frac{\Gamma^{{}^{\prime }}\left(a+b\right)}{\Gamma \left(a+b\right)}, \end{array}$$
$$\begin{array}{c} S\left(X\right)=\log B\left(a,b\right)+\left(1-a\right)\frac{\Gamma^{{}^{\prime }}\left(a\right)}{\Gamma (a)}+\left(1-b\right)\frac{\Gamma^{{}^{\prime }}\left(b\right)}{\Gamma (b)}-\left(2-a-b\right)\frac{\Gamma^{{}^{\prime }}\left(a+b\right)}{\Gamma (a+b)}, \end{array}$$
$$\begin{array}{c} R\left(X\right)=\frac{1}{1-\gamma }\log\frac{B\left(a\gamma -\gamma +1,b\gamma -\gamma +1\right)}{{\left[B(a,b)\right]}^{\gamma }} \end{array}$$
and
$$\begin{array}{c} CE\left(X\right)=-\int_{0}^{\infty }I_{1-x}\left(b,a\right)\log I_{1-x}\left(b,a\right)dx \end{array}$$
for 0<x<1, a>0 and b>0.

43. Inverted beta distribution: for this distribution,
$$\begin{array}{c} f_{X}\left(x\right)=\frac{x^{a-1}{\left(1+x\right)}^{-a-b}}{B(a,b)}, \end{array}$$
$$\begin{array}{c} F_{X}\left(x\right)=I_{\frac{x}{1+x}}\left(a,b\right), \end{array}$$
$$\begin{array}{c} GM\left(X\right)=\frac{\Gamma^{{}^{\prime }}\left(a\right)}{\Gamma \left(a\right)}-\frac{\Gamma^{{}^{\prime }}\left(a+b\right)}{\Gamma \left(a+b\right)}, \end{array}$$
$$\begin{array}{c} S\left(X\right)=\log B\left(a,b\right)+\left(1-a\right)\frac{\Gamma^{{}^{\prime }}\left(a\right)}{\Gamma (a)}+\left(a+b\right)\frac{\Gamma^{{}^{\prime }}\left(a+b\right)}{\Gamma (a+b)}-\left(1+b\right)\frac{\Gamma^{{}^{\prime }}\left(b\right)}{\Gamma (b)}, \end{array}$$
$$\begin{array}{c} R\left(X\right)=\frac{1}{1-\gamma }\log\frac{B\left(a\gamma -\gamma +1,b\gamma +\gamma -1\right)}{{\left[B(a,b)\right]}^{\gamma }} \end{array}$$
and
$$\begin{array}{c} CE\left(X\right)=-\int_{0}^{\infty }I_{\frac{1}{1+x}}\left(b,a\right)\log I_{\frac{1}{1+x}}\left(b,a\right)dx \end{array}$$
for 0<x<1, a>0 and b>0.

44. Kumaraswamy distribution [39]: for this distribution,
$$\begin{array}{c} f_{X}\left(x\right)=abx^{a-1}{\left(1-x^{a}\right)}^{b-1}, \end{array}$$
$$\begin{array}{c} F_{X}\left(x\right)=1-{\left(1-x^{a}\right)}^{b}, \end{array}$$
$$\begin{array}{c} GM\left(X\right)=b\Gamma^{{}^{\prime }}\left(1\right)-b\frac{\Gamma^{{}^{\prime }}\left(1+\frac{1}{a}\right)}{\Gamma \left(1+\frac{1}{a}\right)}, \end{array}$$
$$\begin{array}{c} S\left(X\right)=1-\frac{1}{b}-\log\left(ab\right)+\left(1-a\right)b\Gamma^{{}^{\prime }}\left(1\right)-\left(1-a\right)b\frac{\Gamma^{{}^{\prime }}\left(\frac{1}{a}+1\right)}{\Gamma \left(\frac{1}{a}+1\right)}, \end{array}$$
$$\begin{array}{c} R\left(X\right)=-\log\left(ab\right)+\frac{\gamma \log b}{1-\gamma }+\frac{1}{1-\gamma }\log B\left(\gamma +\frac{1-\gamma }{a},b\gamma +1-\gamma \right) \end{array}$$
and
$$\begin{array}{c} CE\left(X\right)=\frac{b}{a}B\left(\frac{1}{a},b+1\right)\left[\psi \left(b+1\right)-\psi \left(\frac{1}{a}+b+1\right)\right] \end{array}$$
for 0<x<1, a>0 and b>0.

45. Inverted Kumaraswamy distribution [40]: for this distribution,
$$\begin{array}{c} f_{X}\left(x\right)=ab{\left(1+x\right)}^{-a-1}{\left[1-{\left(1+x\right)}^{-a}\right]}^{b-1}, \end{array}$$
$$\begin{array}{c} F_{X}\left(x\right)={\left[1-{\left(1+x\right)}^{-a}\right]}^{b}, \end{array}$$
$$\begin{array}{c} GM\left(X\right)=b\int_{0}^{1}\log\left(y^{-\frac{1}{a}}-1\right){\left(1+y\right)}^{b-1}dy, \end{array}$$
$$\begin{array}{c} S\left(X\right)=-\log\left(ab\right)+\left(1+\frac{1}{a}\right)\left[\frac{\Gamma^{{}^{\prime }}\left(b+1\right)}{\Gamma (b+1)}-\Gamma^{{}^{\prime }}\left(1\right)\right]+1-\frac{1}{b}, \end{array}$$
$$\begin{array}{c} R\left(X\right)=-\log a+\frac{\gamma \log b}{1-\gamma }+\frac{1}{1-\gamma }\log B\left(\gamma +\frac{\gamma -1}{a},b\gamma -\gamma +1\right) \end{array}$$
and
$$\begin{array}{c} CE\left(X\right)=\frac{1}{a}\sum_{k=1}^{\infty }\frac{1}{k}B\left(-\frac{1}{a},kb+1\right)-\frac{1}{a}\sum_{k=1}^{\infty }\frac{1}{k}B\left(-\frac{1}{a},kb+b+1\right) \end{array}$$
for x>0, a>0 and b>0.

46. Normal distribution: for this distribution,
$$\begin{array}{c} f_{X}\left(x\right)=\frac{1}{\sqrt{2\pi }a}\exp\left[-\frac{{\left(x-b\right)}^{2}}{2a^{2}}\right], \end{array}$$
$$\begin{array}{c} F_{X}\left(x\right)=\Phi \left(\frac{x-b}{a}\right), \end{array}$$
$$\begin{array}{c} GM\left(X\right)=\frac{1}{\sqrt{2\pi }a}\int_{-\infty }^{\infty }\log x\exp\left[-\frac{{\left(x-b\right)}^{2}}{2a^{2}}\right]dx, \end{array}$$
$$\begin{array}{c} S\left(X\right)=\log\left(\sqrt{2\pi }a\right)+\frac{1}{2}, \end{array}$$
$$\begin{array}{c} R\left(X\right)=\log\left(\sqrt{2\pi }a\right)-\frac{\log\gamma }{2(1-\gamma )} \end{array}$$
and
$$\begin{array}{c} CE\left(X\right)=-\frac{\sqrt{2}a}{4}\int_{-\infty }^{\infty }\mathrm{erfc}\left(x\right)\log\left[\mathrm{erfc}(x)\right]dx \end{array}$$
for −∞<x<∞, a>0 and −∞<b<∞.

47. Lognormal distribution: for this distribution,
$$\begin{array}{c} f_{X}\left(x\right)=\frac{1}{\sqrt{2\pi }ax}\exp\left[-\frac{{\left(\log x-b\right)}^{2}}{2a^{2}}\right], \end{array}$$
$$\begin{array}{c} F_{X}\left(x\right)=\Phi \left(\frac{\log x-b}{a}\right), \end{array}$$
$$\begin{array}{c} GM(X)=b, \end{array}$$
$$\begin{array}{c} S\left(X\right)=\log\left(\sqrt{2\pi }a\right)+b+\frac{1}{2}, \end{array}$$
$$\begin{array}{c} R\left(X\right)=\sqrt{2\pi }a\exp\left(b\right)+\frac{\left(1-\gamma \right)a^{2}}{2\gamma }-\frac{\log\gamma }{2(1-\gamma )} \end{array}$$
and
$$\begin{array}{c} CE\left(X\right)=-\int_{0}^{\infty }\Phi \left(\frac{b-\log x}{a}\right)\log\left[\Phi \left(\frac{b-\log x}{a}\right)\right]dx \end{array}$$
for x>0, a>0 and −∞<b<∞.

48. Half normal distribution: for this distribution,
$$\begin{array}{c} f_{X}\left(x\right)=\frac{\sqrt{2}}{\sqrt{\pi }a}\exp\left(-\frac{x^{2}}{2a^{2}}\right), \end{array}$$
$$\begin{array}{c} F_{X}\left(x\right)=\mathrm{erf}\left(\frac{x}{\sqrt{2}a}\right), \end{array}$$
$$\begin{array}{c} GM\left(X\right)=\frac{\log2}{2}+\log a+\frac{1}{\sqrt{\pi }}\Gamma^{{}^{\prime }}\left(\frac{1}{2}\right), \end{array}$$
$$\begin{array}{c} S\left(X\right)=\frac{1}{2}-\frac{1}{2}\log\frac{2}{\pi }+\log a, \end{array}$$
$$\begin{array}{c} R\left(X\right)=a\sqrt{\frac{2}{\pi }}-\frac{1}{2}\frac{\log\gamma }{1-\gamma } \end{array}$$
and
$$\begin{array}{c} CE\left(X\right)=-\sqrt{2}a\int_{-\infty }^{\infty }\mathrm{erf}\left(x\right)\log\left[\mathrm{erf}(x)\right]dx \end{array}$$
for x>0 and a>0.

49. Student’s *t* distribution [41]: for this distribution,
$$\begin{array}{c} f_{X}\left(x\right)=\frac{\Gamma \left(\frac{a+1}{2}\right)}{\sqrt{a\pi }\Gamma \left(\frac{a}{2}\right)}{\left(1+\frac{x^{2}}{a}\right)}^{-\frac{a+1}{2}}, \end{array}$$
$$\begin{array}{c} F_{X}\left(x\right)=\frac{1}{2}-\frac{x\Gamma \left(\frac{a+1}{2}\right)}{\sqrt{a\pi }\Gamma \left(\frac{a}{2}\right)}\;{}_{2}F_{1}\left(\frac{1}{2},\frac{a+1}{2};\frac{3}{2};-\frac{x^{2}}{a}\right), \end{array}$$
$$\begin{array}{c} GM\left(X\right)=\frac{\Gamma \left(\frac{a+1}{2}\right)}{\sqrt{a\pi }\Gamma \left(\frac{a}{2}\right)}\int_{-\infty }^{\infty }\log x{\left(1+\frac{x^{2}}{a}\right)}^{-\frac{a+1}{2}}dx, \end{array}$$
$$\begin{array}{c} S\left(X\right)=\log\frac{\sqrt{a\pi }\Gamma \left(\frac{a}{2}\right)}{\Gamma \left(\frac{a+1}{2}\right)}+\frac{\sqrt{a}\left(a+1\right)}{2}\left[\frac{\Gamma^{{}^{\prime }}\left(\frac{a+1}{2}\right)}{\Gamma \left(\frac{a+1}{2}\right)}-\frac{\Gamma^{{}^{\prime }}\left(\frac{a}{2}\right)}{\Gamma \left(\frac{a}{2}\right)}\right], \end{array}$$
$$\begin{array}{c} R\left(X\right)=\frac{1}{1-\gamma }\log\left\{2{\left[\frac{\Gamma \left(\frac{a+1}{2}\right)}{\sqrt{a\pi }\Gamma \left(\frac{a}{2}\right)}\right]}^{\gamma }B\left(\frac{a\gamma +\gamma -1}{2},\frac{1}{2}\right)\right\} \end{array}$$
and
$$\begin{array}{c} CE\left(X\right)=-\int_{-\infty }^{\infty }\left[\frac{1}{2}+\frac{x\Gamma \left(\frac{a+1}{2}\right)}{\sqrt{a\pi }\Gamma \left(\frac{a}{2}\right)}\;{}_{2}F_{1}\left(\frac{1}{2},\frac{a+1}{2};\frac{3}{2};-\frac{x^{2}}{a}\right)\right]\log\left[\frac{1}{2}+\frac{x\Gamma \left(\frac{a+1}{2}\right)}{\sqrt{a\pi }\Gamma \left(\frac{a}{2}\right)}\;{}_{2}F_{1}\left(\frac{1}{2},\frac{a+1}{2};\frac{3}{2};-\frac{x^{2}}{a}\right)\right]dx \end{array}$$
for −∞<x<∞ and a>0.

50. Cauchy distribution: for this distribution,
$$\begin{array}{c} f_{X}\left(x\right)=\frac{1}{\pi \left(1+x^{2}\right)}, \end{array}$$
$$\begin{array}{c} F_{X}\left(x\right)=\frac{1}{2}+\frac{1}{\pi }\arctan\left(x\right), \end{array}$$
$$\begin{array}{c} GM\left(X\right)=\frac{1}{\pi }\int_{-\infty }^{\infty }\frac{\log x}{1+x^{2}}dx, \end{array}$$
$$\begin{array}{c} S\left(X\right)=\log\pi +\Gamma^{{}^{\prime }}\left(1\right)-\frac{\Gamma^{{}^{\prime }}\left(\frac{1}{2}\right)}{\sqrt{\pi }}, \end{array}$$
$$\begin{array}{c} R\left(X\right)=\frac{1}{1-\gamma }\log\left[\frac{2}{\pi^{\gamma }}B\left(\gamma -\frac{1}{2},\frac{1}{2}\right)\right] \end{array}$$
and
$$\begin{array}{c} CE\left(X\right)=-\int_{-\infty }^{\infty }\left(\frac{1}{2}-\frac{1}{\pi }\arctan\left(x\right)\right)\log\left[\frac{1}{2}-\frac{1}{\pi }\arctan\left(x\right)\right]dx \end{array}$$
for −∞<x<∞.

51. Laplace distribution [42]: for this distribution,
$$\begin{array}{c} f_{X}\left(x\right)=\frac{1}{2a}\exp\left(-\frac{|x-b|}{a}\right), \end{array}$$
$$\begin{array}{c} F_{X}\left(x\right)=\left\{\begin{array}{cc} \frac{1}{2}\exp\left(\frac{x-b}{a}\right), & \mathrm{if}\;x\leq b,\\ 1-\frac{1}{2}\exp\left(-\frac{x-b}{a}\right), & \mathrm{if}\;x\geq b, \end{array}\right. \end{array}$$
$$\begin{array}{c} GM\left(X\right)=\frac{1}{2a}\int_{-\infty }^{\infty }\log x\exp\left(-\frac{|x-b|}{a}\right)dx, \end{array}$$
$$\begin{array}{c} S(X)=1+\log(2a), \end{array}$$
$$\begin{array}{c} R\left(X\right)=\log\left(2a\right)-\frac{\log\gamma }{1-\gamma } \end{array}$$
and
$$\begin{array}{c} CE\left(X\right)=\frac{a}{2}+\frac{a}{2}\log2-a{\left.\frac{\partial }{\partial \alpha }B_{\frac{1}{2}}\left(0,2+\alpha \right)\right|}_{\alpha =0} \end{array}$$
for −∞<x<∞, a>0 and −∞<b<∞.

52. Logistic distribution of type I: for this distribution,
$$\begin{array}{c} f_{X}\left(x\right)=\frac{ac\exp(-cx)}{{\left[1+\exp(-cx)\right]}^{a+1}}, \end{array}$$
$$\begin{array}{c} F_{X}\left(x\right)=\frac{1}{{\left[1+\exp(-cx)\right]}^{a}}, \end{array}$$
$$\begin{array}{c} GM\left(X\right)=ac\int_{-\infty }^{\infty }\log x\frac{\exp(-cx)}{{\left[1+\exp(-cx)\right]}^{a+1}}dx, \end{array}$$
$$\begin{array}{c} S\left(X\right)=-\log a-\log c+\left(a+1\right)\psi \left(a+1\right)-a\psi \left(a\right)-\Gamma^{{}^{\prime }}\left(1\right), \end{array}$$
$$\begin{array}{c} R\left(X\right)=-\log c+\frac{1}{1-\gamma }\log\left[a^{\gamma }B\left(a\gamma ,\gamma \right)\right] \end{array}$$
and
$$\begin{array}{c} CE\left(X\right)=-\int_{-\infty }^{\infty }\left\{1-\frac{1}{{\left[1+\exp(-cx)\right]}^{a}}\right\}\log\left\{1-\frac{1}{{\left[1+\exp(-cx)\right]}^{a}}\right\}dx \end{array}$$
for −∞<x<∞, a>0 and c>0.

53. Logistic distribution of type II: for this distribution,
$$\begin{array}{c} f_{X}\left(x\right)=\frac{ac\exp(-acx)}{{\left[1+\exp(-cx)\right]}^{a+1}}, \end{array}$$
$$\begin{array}{c} F_{X}\left(x\right)=1-\frac{1}{{\left[1+\exp(cx)\right]}^{a}}, \end{array}$$
$$\begin{array}{c} GM\left(X\right)=ac\int_{-\infty }^{\infty }\log x\frac{\exp(-acx)}{{\left[1+\exp(-cx)\right]}^{a+1}}dx, \end{array}$$
$$\begin{array}{c} S\left(X\right)=-\log a-\log c+\left(a+1\right)\psi \left(a+1\right)-\Gamma^{{}^{\prime }}\left(1\right)-a\psi \left(a\right), \end{array}$$
$$\begin{array}{c} R\left(X\right)=-\log c+\frac{1}{1-\gamma }\log\left[a^{\gamma }B\left(\gamma ,a\gamma \right)\right] \end{array}$$
and
$$\begin{array}{c} CE\left(X\right)=a\int_{-\infty }^{\infty }{\left[1+\exp(-cx)\right]}^{-a}\log\left[1+\exp(-cx)\right]dx \end{array}$$
for −∞<x<∞, a>0 and c>0.

54. Logistic distribution of type III: for this distribution,
$$\begin{array}{c} f_{X}\left(x\right)=\frac{c\exp(-acx)}{B\left(a,a\right){\left[1+\exp(-cx)\right]}^{2a}}, \end{array}$$
$$\begin{array}{c} F_{X}\left(x\right)=I_{\frac{1}{1+\exp(-cx)}}\left(a,a\right), \end{array}$$
$$\begin{array}{c} GM\left(X\right)=\frac{c}{B(a,a)}\int_{-\infty }^{\infty }\log x\frac{\exp(-acx)}{{\left[1+\exp(-cx)\right]}^{2a}}dx, \end{array}$$
$$\begin{array}{c} S(X)=\log B(a,a)-\log c+2a\psi (2a)-2a\psi (a), \end{array}$$
$$\begin{array}{c} R\left(X\right)=-\log c+\frac{1}{1-\gamma }\log\frac{B\left(a\gamma ,a\gamma \right)}{{\left[B(a,a)\right]}^{\gamma }} \end{array}$$
and
$$\begin{array}{c} CE\left(X\right)=-\int_{-\infty }^{\infty }I_{\frac{\exp(-cx)}{1+\exp(-cx)}}\left(a,a\right)\log I_{\frac{\exp(-cx)}{1+\exp(-cx)}}\left(a,a\right)dx \end{array}$$
for −∞<x<∞, a>0 and c>0.

55. Logistic distribution of type IV [43]: for this distribution,
$$\begin{array}{c} f_{X}\left(x\right)=\frac{c\exp(-bcx)}{B\left(a,b\right){\left[1+\exp(-cx)\right]}^{a+b}}, \end{array}$$
$$\begin{array}{c} F_{X}\left(x\right)=I_{\frac{1}{1+\exp(-cx)}}\left(a,b\right), \end{array}$$
$$\begin{array}{c} GM\left(X\right)=\frac{c}{B(a,b)}\int_{-\infty }^{\infty }\log x\frac{\exp(-bcx)}{{\left[1+\exp(-cx)\right]}^{a+b}}dx, \end{array}$$
$$\begin{array}{c} S(X)=\log B(a,b)-\log c+(a+b)\psi (a+b)-a\psi (a)-b\psi (b), \end{array}$$
$$\begin{array}{c} R\left(X\right)=-\log c+\frac{1}{1-\gamma }\log\frac{B\left(a\gamma ,b\gamma \right)}{{\left[B(a,b)\right]}^{\gamma }} \end{array}$$
and
$$\begin{array}{c} CE\left(X\right)=-\int_{-\infty }^{\infty }I_{\frac{\exp(-cx)}{1+\exp(-cx)}}\left(b,a\right)\log I_{\frac{\exp(-cx)}{1+\exp(-cx)}}\left(b,a\right)dx \end{array}$$
for −∞<x<∞, a>0, b>0 and c>0.

56. Burr distribution [44]: for this distribution,
$$\begin{array}{c} f_{X}\left(x\right)=ckx^{c-1}{\left(1+x^{c}\right)}^{-k-1}, \end{array}$$
$$\begin{array}{c} F_{X}\left(x\right)=1-{\left(1+x^{c}\right)}^{-k}, \end{array}$$
$$\begin{array}{c} GM\left(X\right)=\frac{\Gamma^{{}^{\prime }}\left(1\right)\Gamma \left(k\right)-\Gamma^{{}^{\prime }}\left(k\right)}{c\Gamma (k)}, \end{array}$$
$$\begin{array}{c} S\left(X\right)=1+\frac{1}{k}-\log\left(ck\right)+\frac{1-c}{c\Gamma (k)}\left[\Gamma^{{}^{\prime }}\left(1\right)\Gamma \left(k\right)-\Gamma^{{}^{\prime }}\left(k\right)\right], \end{array}$$
$$\begin{array}{c} R\left(X\right)=\frac{\gamma \log(ck)}{1-\gamma }+\frac{1}{1-\gamma }\log B\left(k\gamma +\frac{\gamma -1}{c},\gamma +\frac{1-\gamma }{c}\right) \end{array}$$
and
$$\begin{array}{c} CE\left(X\right)=-\frac{k}{c}B\left(\frac{1}{c},k-\frac{1}{c}\right)\left[\psi \left(\frac{1}{c}\right)-\psi \left(k\right)\right] \end{array}$$
for x>0, k>0 and c>0.

57. Dagum distribution [45]: for this distribution,
$$\begin{array}{c} f_{X}\left(x\right)=apx^{ap-1}{\left(1+x^{a}\right)}^{-p-1}, \end{array}$$
$$\begin{array}{c} F_{X}\left(x\right)={\left(1+x^{a}\right)}^{-p}, \end{array}$$
$$\begin{array}{c} GM\left(X\right)=-\frac{\Gamma^{{}^{\prime }}\left(1\right)}{a}+\frac{\Gamma^{{}^{\prime }}\left(p\right)}{a\Gamma (p)}, \end{array}$$
$$\begin{array}{c} S\left(X\right)=-\log\left(ap\right)+\left(p+1\right)\left[\Gamma^{{}^{\prime }}\left(1\right)-\frac{\Gamma^{{}^{\prime }}\left(p+1\right)}{\Gamma (p+1)}\right]+\left(\frac{1}{a}-p\right)\left[-\Gamma^{{}^{\prime }}\left(1\right)+\frac{\Gamma^{{}^{\prime }}\left(p\right)}{\Gamma (p)}\right], \end{array}$$
$$\begin{array}{c} R\left(X\right)=\frac{1}{1-\gamma }\log\left[a^{\gamma -1}p^{\gamma }B\left(\frac{\gamma -1}{a},p\gamma +\frac{1-\gamma }{a}\right)\right] \end{array}$$
and
$$\begin{array}{c} CE\left(X\right)=\sum_{k=1}^{\infty }\frac{1}{k}B\left(pk-\frac{1}{a},\frac{1}{a}\right)-\sum_{k=1}^{\infty }\frac{1}{k}B\left(pk+p-\frac{1}{a},\frac{1}{a}\right) \end{array}$$
for x>0, a>0 and p>0.

58. *J* shaped distribution [46]: for this distribution,
$$\begin{array}{c} f_{X}\left(x\right)=2a\left(1-x\right){\left[x(2-x)\right]}^{a-1}, \end{array}$$
$$\begin{array}{c} F_{X}\left(x\right)={\left[x(2-x)\right]}^{a}, \end{array}$$
$$\begin{array}{c} GM\left(X\right)=a2^{a}{\left.\frac{\partial }{\partial \alpha }\left[\frac{1}{a+\alpha }\;{}_{2}F_{1}\left(a+\alpha ,1-a;a+\alpha +1;\frac{1}{2}\right)\right]\right|}_{\alpha =0}, \end{array}$$
$$\begin{array}{ccc} & & S\left(X\right)=-\log\left(2a\right)+a\left(1-a\right)2^{a}{\left.\frac{\partial }{\partial \alpha }\left[\frac{1}{a+\alpha }\;{}_{2}F_{1}\left(a+\alpha ,1-a;a+\alpha +1;\frac{1}{2}\right)\right]\right|}_{\alpha =0}\\ & & \;\;-a2^{a}{\left.\frac{\partial }{\partial \alpha }\left[B\left(a,\alpha +2\right)\;{}_{2}F_{1}\left(a,1-a;a+\alpha +2;\frac{1}{2}\right)\right]\right|}_{\alpha =0}\\ & & \;\;+\frac{2^{a}\left(1-a\right)}{1+a}{\left.\frac{\partial }{\partial \alpha }\left[2^{\alpha }\;{}_{2}F_{1}\left(a,1-\alpha -a;a+1;\frac{1}{2}\right)\right]\right|}_{\alpha =0}, \end{array}$$
$$\begin{array}{c} R\left(X\right)=\frac{1}{1-\gamma }\log\left[a^{\gamma }2^{a\gamma }B\left(a\gamma -\gamma +1,\gamma +1\right)\;{}_{2}F_{1}\left(a\gamma -\gamma +1,\gamma -a\gamma ;a\gamma +2;\frac{1}{2}\right)\right] \end{array}$$
and
$$\begin{array}{ccc} & & CE\left(X\right)=\sum_{k=1}^{\infty }\frac{2^{ak}}{k(ak+1)}\;{}_{2}F_{1}\left(ak+1,-ak;ak+2;\frac{1}{2}\right)\\ & & \;\;-\sum_{k=1}^{\infty }\frac{2^{a(k+1)}}{k(ak+a+1)}\;{}_{2}F_{1}\left(ak+a+1,-ak-a;ak+a+2;\frac{1}{2}\right) \end{array}$$
for 0<x<1 and a>0.

59. Nadarajah–Haghighi distribution [47]: for this distribution,
$$\begin{array}{c} f_{X}\left(x\right)=ab{\left(1+bx\right)}^{a-1}\exp\left[1-{\left(1+bx\right)}^{a}\right], \end{array}$$
$$\begin{array}{c} F_{X}\left(x\right)=1-\exp\left[1-{\left(1+bx\right)}^{a}\right], \end{array}$$
$$\begin{array}{c} GM\left(X\right)=-e\log b+e\int_{0}^{\infty }\log\left(y^{\frac{1}{a}}-1\right)\exp\left(-y\right)dy, \end{array}$$
$$\begin{array}{c} S\left(X\right)=-\log\left(ab\right)+\left(1-a\right)e{\left.\frac{\partial }{\partial \alpha }\Gamma \left(\frac{\alpha }{a}+1,1\right)\right|}_{\alpha =0}+1, \end{array}$$
$$\begin{array}{c} R\left(X\right)=\frac{\gamma }{1-\gamma }-\frac{\log(ab)}{1-\gamma }-\left(\gamma +\frac{1-\gamma }{a}\right)\frac{\log\gamma }{1-\gamma }+\frac{1}{1-\gamma }\log\Gamma \left(\gamma +\frac{1-\gamma }{a},\gamma \right) \end{array}$$
and
$$\begin{array}{c} CE\left(X\right)=\frac{1}{b}+\frac{e}{b}\left(\frac{1}{a}-1\right)\Gamma \left(\frac{1}{a}+1,1\right) \end{array}$$
for x>0, a>0 and b>0.

60. Two-sided power distribution [48]: for this distribution,
$$\begin{array}{c} f_{X}\left(x\right)=\left\{\begin{array}{cc} a{\left(\frac{x}{\theta }\right)}^{a-1}, & \mathrm{if}\;0<x\leq \theta ,\\ a{\left(\frac{1-x}{1-\theta }\right)}^{a-1}, & \mathrm{if}\;\theta \leq x<1, \end{array}\right. \end{array}$$
$$\begin{array}{c} F_{X}\left(x\right)=\left\{\begin{array}{cc} \theta {\left(\frac{x}{\theta }\right)}^{a}, & \mathrm{if}\;0<x\leq \theta ,\\ 1-\left(1-\theta \right){\left(\frac{1-x}{1-\theta }\right)}^{a}, & \mathrm{if}\;\theta \leq x<1, \end{array}\right. \end{array}$$
$$\begin{array}{c} GM\left(X\right)=\theta \log\theta -\frac{\theta }{a}+\frac{a}{{\left(1-\theta \right)}^{a-1}}{\left.\frac{\partial }{\partial \alpha }B_{1-\theta }\left(a,\alpha +1\right)\right|}_{\alpha =0}, \end{array}$$
$$\begin{array}{c} S\left(X\right)=-\log a-\frac{1-a}{a}, \end{array}$$
$$\begin{array}{c} R\left(X\right)=\frac{1}{1-\gamma }\log\frac{a^{\gamma }}{a\gamma -\gamma +1} \end{array}$$
and
CE(X)=∑k=1∞θk+1k(kθ+1)−∑k=1∞1k∑m=0kkm(−1)m(1−θ)m+1am+1−∑k=1∞θk+2k(kθ+k+1)+∑k=1∞1k∑m=0k+1k+1m(−1)m(1−θ)m+1am+1
for a>0.

61. Power Lindley distribution [49]: for this distribution,
$$\begin{array}{c} f_{X}\left(x\right)=\frac{ab^{2}}{b+1}\left(1+x^{a}\right)x^{a-1}\exp\left(-bx^{a}\right), \end{array}$$
$$\begin{array}{c} F_{X}\left(x\right)=1-\left(1+\frac{bx^{a}}{b+1}\right)\exp\left(-bx^{a}\right), \end{array}$$
$$\begin{array}{c} GM\left(X\right)=\frac{b\Gamma^{{}^{\prime }}\left(1\right)+\Gamma^{{}^{\prime }}\left(2\right)}{a(b+1)}-\frac{\log b}{a}, \end{array}$$
$$\begin{array}{c} S\left(X\right)=-\log\frac{ab}{b+1}-\frac{\log b}{a}+\frac{b(1-a)}{a(b+1)}\left[\Gamma^{{}^{\prime }}\left(1\right)+\frac{1}{b}\Gamma^{{}^{\prime }}\left(2\right)\right]+\frac{b+2}{b+1}-\frac{\exp(b)}{b+1}{\left.\frac{\partial }{\partial \alpha }\Gamma \left(\alpha +2,b\right)\right|}_{\alpha =0}, \end{array}$$
$$\begin{array}{ccc} & & R\left(X\right)=-\log a-\frac{\gamma }{1-\gamma }\log\left(b+1\right)\\ & & \;\;+\frac{1}{1-\gamma }\log\left[\Gamma \left(\gamma +\frac{1-\gamma }{a}\right)\Psi \left(\gamma +\frac{1-\gamma }{a},\frac{1-\gamma }{a}+1;b\gamma \right)\right] \end{array}$$
and
$$\begin{array}{ccc} & & CE\left(X\right)=\frac{1}{ab^{\frac{1}{a}}}\left[\Gamma \left(\frac{1}{a}+1\right)+\frac{1}{b+1}\Gamma \left(\frac{1}{a}+2\right)\right]\\ & & \;\;-\frac{\Gamma \left(\frac{1}{a}\right){\left(1+b\right)}^{\frac{1}{a}}}{ab^{\frac{1}{a}}}{\left.\frac{\partial }{\partial \alpha }\Psi \left(\frac{1}{a},\frac{1}{a}+\alpha +2;b+1\right)\right|}_{\alpha =0} \end{array}$$
for x>0, a>0 and b>0.

62. Modified slash Lindley–Weibull distribution [50]: for this distribution,
$$\begin{array}{c} f_{X}\left(x\right)=\frac{2a^{3}b^{2}x^{a-1}}{a+1}\frac{\left(a+2\right)x^{a}+2b^{a}}{{\left(ax^{a}+2b^{a}\right)}^{3}}, \end{array}$$
$$\begin{array}{c} F_{X}\left(x\right)=\frac{a^{2}x^{a}}{a+1}\frac{\left(a+1\right)x^{a}+2b^{a}}{{\left(ax^{a}+2b^{a}\right)}^{2}}, \end{array}$$
$$\begin{array}{ccc} & & GM\left(X\right)=\frac{(a+2)\log2}{4(a+1)}-\frac{(a+2)\log a}{4(a+1)}+\frac{(a+2)\log b}{2(a+1)}\Gamma^{{}^{\prime }}\left(2\right)-\frac{a+2}{a(a+1)}\Gamma^{{}^{\prime }}\left(1\right)\\ & & \;\;+\frac{b^{1-a}a^{2-\frac{1}{a}}\log2}{2^{\frac{5}{2}}\left(a+1\right)}\Gamma \left(\frac{1}{a}\right)\Gamma \left(3-\frac{1}{a}\right)+\frac{b^{1-a}a^{2-\frac{1}{a}}\log b}{2^{a-\frac{1}{2}}\left(a+1\right)}\Gamma \left(\frac{1}{a}\right)\Gamma \left(3-\frac{1}{a}\right)\\ & & \;\;-\frac{b^{1-a}\sqrt{a}\log2}{2^{\frac{3}{2}}\left(a+1\right)}\Gamma \left(\frac{1}{a}\right)\Gamma \left(3-\frac{1}{a}\right)+\frac{b^{1-a}a^{1-\frac{1}{a}}}{2^{\frac{3}{2}}\left(a+1\right)}\Gamma^{{}^{\prime }}\left(\frac{1}{a}\right)\Gamma \left(3-\frac{1}{a}\right)\\ & & \;\;-\frac{b^{1-a}a^{1-\frac{1}{a}}}{2^{\frac{3}{2}}\left(a+1\right)}\Gamma \left(\frac{1}{a}\right)\Gamma^{{}^{\prime }}\left(3-\frac{1}{a}\right), \end{array}$$
$$\begin{array}{ccc} & & S\left(X\right)=-\log\frac{2a^{3}b^{a}}{a+1}+\frac{(1-a)(a+2)\log2}{4(a+1)}-\frac{(1-a)(a+2)\log a}{4(a+1)}\\ & & \;\;+\frac{(1-a)(a+2)\log b}{2(a+1)}\Gamma^{{}^{\prime }}\left(2\right)-\frac{\left(1-a)(a+2\right)}{a(a+1)}\Gamma^{{}^{\prime }}\left(1\right)\\ & & \;\;+\frac{\left(1-a\right)b^{1-a}a^{2-\frac{1}{a}}\log2}{2^{\frac{5}{2}}\left(a+1\right)}\Gamma \left(\frac{1}{a}\right)\Gamma \left(3-\frac{1}{a}\right)+\frac{\left(1-a\right)b^{1-a}a^{2-\frac{1}{a}}\log b}{2^{a-\frac{1}{2}}\left(a+1\right)}\Gamma \left(\frac{1}{a}\right)\Gamma \left(3-\frac{1}{a}\right)\\ & & \;\;-\frac{\left(1-a\right)b^{1-a}\sqrt{a}\log2}{2^{\frac{3}{2}}\left(a+1\right)}\Gamma \left(\frac{1}{a}\right)\Gamma \left(3-\frac{1}{a}\right)+\frac{\left(1-a\right)b^{1-a}a^{1-\frac{1}{a}}}{2^{\frac{3}{2}}\left(a+1\right)}\Gamma^{{}^{\prime }}\left(\frac{1}{a}\right)\Gamma \left(3-\frac{1}{a}\right)\\ & & \;\;-\frac{\left(1-a\right)b^{1-a}a^{1-\frac{1}{a}}}{2^{\frac{3}{2}}\left(a+1\right)}\Gamma \left(\frac{1}{a}\right)\Gamma^{{}^{\prime }}\left(3-\frac{1}{a}\right)\\ & & \;\;-\frac{2ab^{a}}{a+1}{\left.\frac{\partial }{\partial \alpha }\left[{\left(\frac{a+2}{a}\right)}^{\alpha +1}\int_{0}^{\infty }{\left(y+\frac{2ab^{a}}{a+2}\right)}^{\alpha +1}{\left(y+2b^{a}\right)}^{-3}dy\right]\right|}_{\alpha =0}\\ & & \;\;+\frac{3a}{a+1}\left\{\frac{a+2}{a}\left[1+\log\left(2b^{2}\right)\right]-\frac{a-2}{4}\left[1+2\log\left(2b^{2}\right)\right]\right\}, \end{array}$$
$$\begin{array}{c} R\left(X\right)=\frac{1}{1-\gamma }\log\left[{\left(\frac{2a^{2}b^{a}}{a+1}\right)}^{\gamma }a^{\frac{\gamma -1}{a}-1}\int_{0}^{\infty }y^{\gamma +\frac{1-\gamma }{a}}{\left(\frac{a+2}{a}y+2b^{a}\right)}^{\gamma }{\left(y+2b^{a}\right)}^{-3\gamma }dy\right] \end{array}$$
and
$$\begin{array}{ccc} & & CE\left(X\right)=-\log\left(\frac{2b^{a}}{a+1}\right)\frac{2^{\frac{1}{a}}b}{a^{\frac{1}{a}+2}}\left(a+\frac{1}{a}\right)\Gamma \left(\frac{1}{a}\right)\Gamma \left(1-\frac{1}{a}\right)\\ & & \;\;-\frac{{\left(a+1\right)}^{\frac{1}{a}}{\left(2b^{a}\right)}^{\frac{1}{a}}}{a^{\frac{1}{a}+1}{\left(a+2\right)}^{\frac{1}{a}}}{\left.\frac{\partial }{\partial \alpha }\left\{{\left[2\left(a+1\right)b^{a}\right]}^{\alpha }B\left(\frac{1}{a},1-\alpha -\frac{1}{a}\right)\;{}_{2}F_{1}\left(\frac{1}{a},2;1-\alpha ;\frac{1}{a+2}\right)\right\}\right|}_{\alpha =0}\\ & & \;\;-\frac{2^{\frac{1}{a}+1}b\log2}{a^{\frac{1}{a}+1}}\Gamma \left(\frac{1}{a}\right)\Gamma \left(2-\frac{1}{a}\right)-\frac{2^{\frac{1}{a}+1}b\log b}{a^{\frac{1}{a}}}\Gamma \left(\frac{1}{a}\right)\Gamma \left(2-\frac{1}{a}\right)\\ & & \;\;+\frac{2^{\frac{1}{a}+1}b}{a^{\frac{1}{a}}}\Gamma \left(\frac{1}{a}\right)\Gamma^{{}^{\prime }}\left(2-\frac{1}{a}\right)-\frac{2^{\frac{1}{a}+1}b}{a^{\frac{1}{a}+1}}\Gamma \left(\frac{1}{a}\right)\Gamma \left(2-\frac{1}{a}\right)\Gamma^{{}^{\prime }}\left(2\right)\\ & & \;\;-\frac{2^{\frac{1}{a}+1}b\left(a+2\right)\log2}{a^{\frac{1}{a}+1}\left(a+1\right)}\Gamma \left(1+\frac{1}{a}\right)\Gamma \left(1-\frac{1}{a}\right)\\ & & \;\;-\frac{2^{\frac{1}{a}+1}b\left(a+2\right)\log b}{a^{\frac{1}{a}}\left(a+1\right)}\Gamma \left(1+\frac{1}{a}\right)\Gamma \left(1-\frac{1}{a}\right)\\ & & \;\;+\frac{2^{\frac{1}{a}+1}b\left(a+2\right)}{a^{\frac{1}{a}+1}\left(a+1\right)}\Gamma \left(1+\frac{1}{a}\right)\Gamma^{{}^{\prime }}\left(1-\frac{1}{a}\right)\\ & & \;\;-\frac{2^{\frac{1}{a}+1}b\left(a+2\right)\log2}{a^{\frac{1}{a}+1}\left(a+1\right)}\Gamma \left(\frac{1}{a}\right)\Gamma \left(2-\frac{1}{a}\right)\Gamma^{{}^{\prime }}\left(2\right) \end{array}$$
for x>0, a>0 and b>0.

63. Reciprocal distribution: for this distribution,
$$\begin{array}{c} f_{X}\left(x\right)=\frac{1}{x\left(\log b-\log a\right)}, \end{array}$$
$$\begin{array}{c} F_{X}\left(x\right)=\frac{\log x-\log a}{\log b-\log a}, \end{array}$$
$$\begin{array}{c} GM\left(X\right)=\frac{\log b+\log a}{2}, \end{array}$$
$$\begin{array}{c} S\left(X\right)=-\log\left[\log b-\log a\right]+\frac{\log b+\log a}{2}, \end{array}$$
$$\begin{array}{c} R\left(X\right)=\frac{1}{1-\gamma }\log\left\{\frac{b^{1-\gamma }-a^{1-\gamma }}{1-\gamma }{\left[\log b-\log a\right]}^{\gamma }\right\} \end{array}$$
and
$$\begin{array}{c} CE\left(X\right)=\frac{\left(b-2a\right)\log\left[\log b-\log a\right]}{\log b-\log a}-\frac{b}{\log b-\log a}{\left.\frac{\partial }{\partial \alpha }\gamma \left(\alpha +2,\log b-\log a\right)\right|}_{\alpha =0} \end{array}$$
for 0<a≤x≤b<∞.

## 4. Conclusions

We have derived the most comprehensive collection of explicit expressions for the geometric mean, Shannon entropy, Rényi entropy and the cumulative residual entropy for the following continuous univariate distributions: 1. Gauss hypergeometric beta distribution, 2. *q* Weibull distribution, 3. *q* exponential distribution, 4. Weighted exponential distribution, 5. Teissier distribution, 6. Maxwell distribution, 7. Inverse Maxwell distribution, 8. Power Maxwell distribution, 9. Inverse power Maxwell distribution, 10. Omega distribution, 11. Colak et al.’s distribution, 12. Bimodal beta distribution, 13. Confluent hypergeometric beta distribution, 14. Libby and Novick’s beta distribution, 15. Generalized beta distribution, 16. Log-logistic distribution, 17. Inverse Gaussian distribution, 18. Gompertz distribution, 19. Exponential distribution, 20. Inverse exponential distribution, 21. Exponentiated exponential distribution, 22. Gamma distribution, 23. Chisquare distribution, 24. Chi distribution, 25. Inverse gamma distribution, 26. Inverse chisquare distribution, 27. Inverse chi distribution, 28. Rayleigh distribution, 29. Weibull distribution, 30. Inverse Rayleigh distribution, 31. Inverse Weibull distribution, 32. Gumbel distribution, 33. Generalized extreme value distribution, 34. Generalized gamma distribution, 35. Pareto distribution of type I, 36. Pareto distribution of type II, 37. Generalized Pareto distribution, 38. Uniform distribution, 39. Power function distribution of type I, 40. Power function distribution of type II, 41. Arcsine distribution, 42. Beta distribution, 43. Inverted beta distribution, 44. Kumaraswamy distribution, 45. Inverted Kumaraswamy distribution, 46. Normal distribution, 47. Lognormal distribution, 48. Half normal distribution, 49. Student’s *t* distribution, 50. Cauchy distribution, 51. Laplace distribution, 52. Logistic distribution of type I, 53. Logistic distribution of type II, 54. Logistic distribution of type III, 55. Logistic distribution of type IV, 56. Burr distribution, 57. Dagum distribution, 58. *J* shaped distribution, 59. Nadarajah–Haghighi distribution, 60. Two-sided power distribution, 61. Power Lindley distribution, 62. Modified slash Lindley–Weibull distribution, 63. Reciprocal distribution. This collection could be a useful reference for both theoreticians and practitioners of entropies. Future work will be to derive similar collections of explicit expressions for entropies of discrete univariate distributions, continuous bivariate distributions, discrete bivariate distributions, continuous multivariate distributions, discrete multivariate distributions, continuous matrix variate distributions, discrete matrix variate distributions, continuous complex variate distributions, and discrete complex variate distributions.

## Data Availability

Not applicable.

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
