# Peer review of "Explicit Expressions for Most Common Entropies"

_entropy, 2023, doi:10.3390/e25030534_

Round 1
Reviewer 1 Report
In regards to the paper “Explicit expressions for most common entropies”, the authors present a series of explicit expressions for the most common entropies found in the literature. In general, the paper is well written, however, the topic is not new and the results are poorly discussed, in consequence, this review does not recommend this paper for publication at Entropy, this review recommends that the manuscript is more suitable for a book chapter on related topics. Besides, the paper contains several typos, for instance, several references to the equations contain errors.
Reviewer 2 Report
I will not go ahead wth the revision of this manuscript until the authors solve some basic presentation issues, such as wrong cross-references to equations (lines 35, 64, 67) and undefined symbols: H(X) in lines 25-27, E(X) in eq. (5), etc. It's utterly disrespectful toward editors and reviewers, and potentially towards readers, to submit a manuscript with such elementary problems.
Round 2
Reviewer 1 Report
The authors have addressed all the comments, thus I could recommend the paper for its publication.
Reviewer 2 Report
There is still a presentation problem with this paper: results are very difficult to trace along the text. I suggest that, in section 3, the authors introduce numbering for each distribution they consider, for instance:
1. Gauss hypergeometric beta distribution...
2. q Weibull distribution...
3. q exponential distribution...
etc.
Then, at the end, they should complement the conclusion with a list of all distributions considered in section 3, namely:
1. Gauss hypergeometric beta distribution.
2. q Weibull distribution.
3. q exponential distribution.
etc.
This should help the reader to rapidly locate the results for a specific distribution he/she may be interested in.
Moreover, the formula of the expectation value E(.) just following Eq. (5) should be given explicitly.
